# Neural quantum state study of fracton models

**Marc Machaczek[1,2,3]\*, Lode Pollet[2,3]† and Ke Liu[4,5,2,3]‡**

**1** Center for Electronic Correlations and Magnetism,
University of Augsburg, 86135 Augsburg, Germany
**2** Arnold Sommerfeld Center for Theoretical Physics,
Ludwig-Maximilians-Universität München,
Theresienstr. 37, 80333 München, Germany
**3** Munich Center for Quantum Science and Technology
(MCQST), Schellingstr. 4, 80799 München, Germany
**4** Hefei National Research Center for Physical Sciences
at the Microscale and School of Physical Sciences,
University of Science and Technology of China, Hefei 230026, China
**5** Shanghai Research Center for Quantum Science and CAS Center
for Excellence in Quantum Information and Quantum Physics,
University of Science and Technology of China,Shanghai 201315, China

⋆ marc.machaczek@uni-a.de , † lode.pollet@lmu.de , ‡ ke.liu@ustc.edu.cn

## Abstract

**Fracton models host unconventional topological orders in three and higher dimensions and provide promising candidates for quantum memory platforms. Understanding their robustness against quantum fluctuations is an important task but also poses great challenges due to the lack of efficient numerical tools. In this work, we establish neural quantum states (NQS) as new tools to study phase transitions in these models. Exact and efficient parametrizations are derived for three prototypical fracton codes — the checkerboard and X-cube model, as well as Haah's code — both in terms of a restricted Boltzmann machine (RBM) and a correlation-enhanced RBM. We then adapt the correlation-enhanced RBM architecture to a perturbed checkerboard model and reveal its strong first-order phase transition between the fracton phase and a trivial field-polarizing phase. To this end, we simulate this highly entangled system on lattices of up to 512 qubits with high accuracy, representing a cutting-edge application of variational neural-network methods. In addition, we reproduce the phase transition of the X-cube model previously obtained with quantum Monte Carlo and high-order series expansion methods. Our work demonstrates the remarkable potential of NQS in studying complicated three-dimensional problems and highlights physics-oriented constructions of NQS architectures.**

 Check for updates

## 1  Introduction

Fracton topological orders are novel states of quantum matter that support mobility-constrained quasiparticles with a subextensively divergent ground-state degeneracy [1–10]. Understanding their phase transitions is an important topic and of great interest in condensed matter physics, quantum information, and quantum field theories. The fate of fracton orders against fluctuations ultimately determines their applications as robust topological error-correcting platforms [1, 2, 11–13] for storing and processing quantum information [14, 15]. Moreover, the dynamics and condensation of fracton excitations may reveal new organizing principles beyond the celebrated Landau-Ginzburg-Wilson paradigm [16–20]. Nevertheless, the study of fracton phase transitions suffers from the lack of efficient tools. Established field-

theoretical constructions face fundamental challenges because of the restricted mobility of the quasi-particles and the strong system size dependence of the ground-state degeneracy [21–24]. Numerical simulations of lattice fracton models are in a similarly difficult situation since the strongly entangled fracton topological orders only exist in three and higher dimensions. Furthermore, interaction terms driving a phase transition may trigger the infamous sign problem. Applications of standard techniques, such as exact diagonalization [25, 26], tensor network states [27, 28], and quantum Monte Carlo simulations [29, 30] are hence limited in scope.

NQS represent a modern technique for addressing the ground state problem of strongly correlated systems [31] that is not inherently limited by dimensionality or the sign problem [32, 33]: The quantum wave function is parameterized in terms of a neural network, thereby avoiding the exponential growth of the Hilbert space by restricting the ground state search to a smaller parameterized subspace. Observables and gradients can then be estimated using Monte-Carlo sampling, which places NQS in the variational Monte Carlo (VMC) [34] framework.

The restricted Boltzmann machine (RBM) is the prototypical shallow NQS architecture [31, 35–37] and enjoys much popularity due to its relation with statistical physics, simple closed-form expressions, and connection to tensor network states [38, 39]. Although previous works have shown the RBMs' ability to represent highly entangled systems and topological orders [33, 40–43], they typically work in low dimensions or rely on the exactly solvable limit of topological stabilizer codes.

The objectives of this work are twofold: Firstly, we explore the potential of neural quantum states (NQS) in representing three-dimensional (3D) lattice Hamiltonians with long-range entanglement. In particular, we construct efficient NQS parametrizations for the checkerboard model, the X-cube model, and Haah's code [1, 2]. Secondly, we study the phase transitions of fracton models subject to external fields. Since the phase transitions of the X-cube model and Haah's code under uniform magnetic fields have previously been investigated with quantum Monte Carlo simulations [44, 45] and series expansions [46], we focus our simulations on the checkerboard model. They are found to be strongly first order, just as was the case for the X-cube model and Haah's code. It is an open question whether RBMs are powerful enough in dealing with such 3D problems: The universal approximation theorem [47–49] does not guarantee a favorable scaling in the number of parameters when increasing the system size or the desired accuracy, and we find that regular RBMs and feedforward neural networks (FFNNs) have difficulties in learning the ground state of small 3D topological codes. The correlation-enhanced RBM, by contrast, performs remarkably well. Our results thus complement our knowledge of fracton phase transitions and demonstrate the power of NQS in studying highly entangled 3D systems.

This paper is organized as follows: In Section 2, we define the Hamiltonians of the three prototypical fracton models. In Section 3, we review the NQS method and the network architectures used in this work, in particular, a correlation-enhanced RBM (cRBM) introduced in Ref. [50]. This cRBM architecture is used to produce the main results of this paper, in particular to simulate the perturbed checkerboard model on large lattices. Exact representations of unperturbed fracton models are constructed in terms of both regular RBMs and cRBMs in Section 4. Section 5 is devoted to an overview of the NQS simulations, including the key aspects of hyperparameters, GPU parallelization, as well as training and sampling strategies. A comprehensive benchmark on a small lattice is followed in Section 6, where we perturb the checkerboard model by three different field directions including a $\sigma_y$ coupling. Section 7 discusses the field-induced phase transitions by focusing on the $\sigma_x$-perturbation, as the checkerboard model is self-dual under swapping $\sigma_x$ and $\sigma_z$. We also discuss the problem and possible solutions for scaling up simulations in the presence of an imaginary $\sigma_y$ term. We conclude in Section 8 with a discussion.

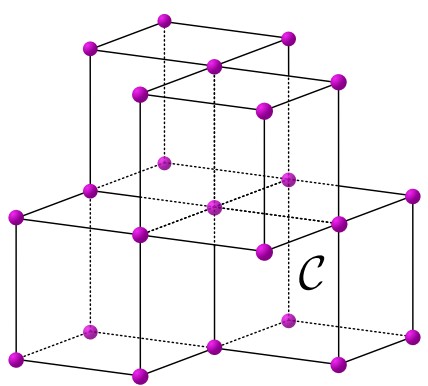

(a) Checkerboard model.

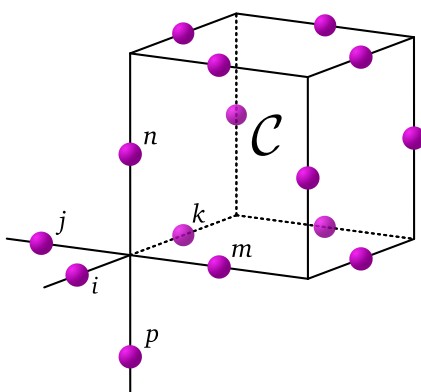

(b) X-cube model.

Figure 1: The positions of the qubits are indicated by the purple balls. For the checkerboard model 1a, two stabilizer generators $A_{\mathcal{C}}$ and $B_{\mathcal{C}}$ are assigned to every second cube $\mathcal{C}$. For the X-cube model 1b, a stabilizer generator $A_{\mathcal{C}}$ is assigned to every cube, and three generators $B_{\mathcal{S}_\mu}$ are assigned to every vertex.

In addition, we provide extensive details of our NQS implementations in the Appendices A–D. We expect them to serve as a technical guide for readers who are not yet familiar with NQS simulations but are willing to explore these methods. The complete code with a detailed documentation and a toy-model example of the 2D toric code is available in [51].

## 2 Fracton models

In this section, we introduce the three prototypical fracton models [1,2] — the checkerboard model, the X-cube model, and Haah's code, and review their basic properties.

### 2.1 Checkerboard model

The checkerboard model is defined on a three-dimensional cubic lattice where a single qubit is placed on every vertex. Denoting the three Pauli matrices by $\sigma^x, \sigma^y$ and $\sigma^z$, the Hamiltonian of the checkerboard model reads

$$H_{\mathrm{CB}} = -\sum_{\mathcal{C}} A_{\mathcal{C}} - \sum_{\mathcal{C}} B_{\mathcal{C}}, \text{ with } A_{\mathcal{C}} := \prod_{i \in \mathcal{C}} \sigma_i^x, \ B_{\mathcal{C}} := \prod_{i \in \mathcal{C}} \sigma_i^z. \tag{1}$$

The sums include every second cube $\mathcal{C}$ of the cubic lattice, corresponding to just one color of a three-dimensional checkerboard. For the $A_{\mathcal{C}}$ and $B_{\mathcal{C}}$ operators, the tensor product is taken over the 8 corners (or vertices) of each such cube. This is illustrated in Figure 1a.

Being a stabilizer code, its so-called stabilizer generators $A_{\mathcal{C}}$ and $B_{\mathcal{C}}$ mutually commute, which is due to any two cubes sharing either none, two or all vertices. Hence, the Hamiltonian 1 can be solved exactly and the ground state manifold is determined by the conditions $A_{\mathcal{C}} = B_{\mathcal{C}} = +1 \ \forall \mathcal{C}$. Using the projectors $\frac{1+A_{\mathcal{C}}}{2}$, it is possible to write one ground state as

$$|\mathrm{GS}_{\mathrm{CB}}\rangle \propto \prod_{\mathcal{C}} \frac{(1+A_{\mathcal{C}})}{2} |\uparrow \ldots \uparrow\rangle, \tag{2}$$

where $|\uparrow \ldots \uparrow\rangle$ denotes the all-spin-up state in the $z$-basis. If periodic boundary conditions are imposed on the lattice, the ground state degeneracy of the checkerboard model scales

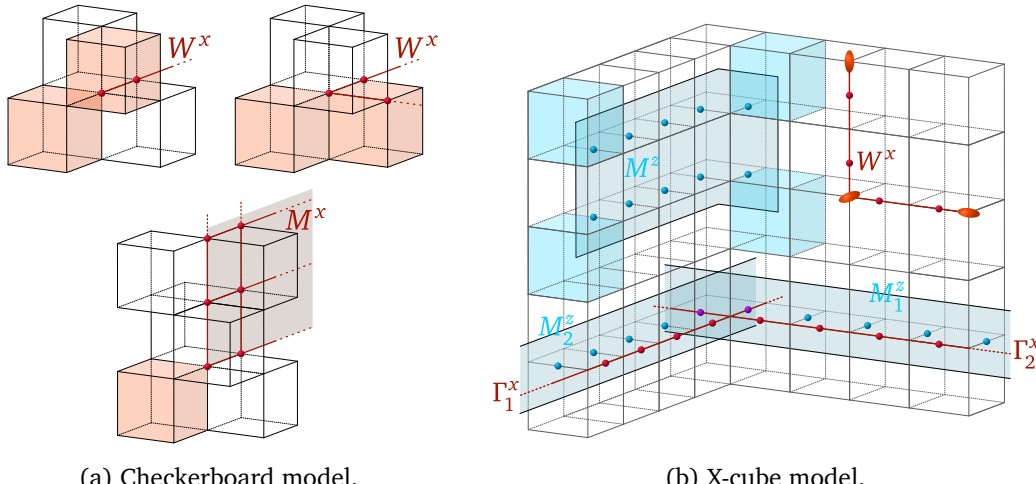

(a) Checkerboard model.        (b) X-cube model.

Figure 2: 2a shows the elementary excitations in the checkerboard model. The string operators $W^x$ and membrane operators $M^x$ act with $\sigma^x$ on the sites indicated by the red dots. The violated stabilizer generators with $B_{\mathcal{C}} = -1$ corresponding to the immobile fracton excitations are indicated by the red colored cubes. The situation is symmetric under $x \leftrightarrow z$. 2b shows the elementary excitations in the X-cube model and the logical operators acting on its code space. Membrane operators $M^z$, which act on the sites indicated by the blue dots via $\sigma^z$, create four spatially separated fractons at their corners. The violated $A_{\mathcal{C}}$ stabilizer generators are indicated by the blue colored cubes. $W^x$ string-like operators, which act with $\sigma^x$ on the sites indicated by the red dots, create lineon excitations at both ends and every corner, which are shown as red ellipsoids. $(M_1^z, \Gamma_1^x)$ and $(M_2^z, \Gamma_2^x)$ are two pairs of logical operators acting on the code space of the X-cube model.

exponentially in linear system size, for instance as $2^{6L-6}$ on a three-torus of size $L \times L \times L$ - a characteristic property of fracton models. Similar to topologically ordered phases, the ground state degeneracy is robust with respect to local perturbations. However, the dependence on system size, a non-topological feature, already indicates that fracton models are not described by conventional topological order [3].

The elementary excitations in the checkerboard model are strictly immobile and correspond to violated stabilizer generators $A_{\mathcal{C}} = -1$ or $B_{\mathcal{C}} = -1$ with an energy cost of $+2$ each. These so-called fractons - the hallmark of fracton models - are created at ends and corners of $\sigma^z$- or $\sigma^x$-type string and membrane operators, as is illustrated in Figure 2a. This implies in particular that they must be created at least in pairs and cannot be moved individually without an additional energy cost. However, such excitations can form composite quasiparticles that are free to move on certain submanifolds. For instance, two fracton excitations at the end of one string operator can move freely as a pair along the direction of the string. The logical operators acting on the ground state manifold, also referred to as the code space of the stabilizer code, correspond to non-contractible line operators $\Gamma^x = \prod \sigma^x$ and $\Gamma^z = \prod \sigma^z$, which act with $\sigma^x$ and $\sigma^z$ along strings that wind around the three-torus, respectively. There are $6L-6$ independent pairs of $(\Gamma^x, \Gamma^z)$ that encode $6L-6$ logical qubits in a fault-tolerant way [2, 52].

## 2.2 X-cube model

For the X-cube model, a qubit is placed on every link of the three-dimensional cubic lattice. Then, we define $A_{\mathcal{C}} := \prod_{i \in \mathcal{C}} \sigma_i^x$ for every cube $\mathcal{C}$, where the tensor product involves all 12 qubits on the edges of the cube. Further, to each vertex $\mathcal{V}$ we assign three different stars:

$\mathcal{S}^x = \{n, j, p, m\}$, $\mathcal{S}^y = \{n, i, p, k\}$, and $\mathcal{S}^z = \{k, j, i, m\}$, the indices of which are marked in Figure 1b. In this notation, $S^\mu$ is the star orthogonal to direction $\mu \in \{x, y, z\}$. Then, for every vertex $\mathcal{V}$ we define the star operators as $B_{\mathcal{S}^\mu} := \prod_{i \in \mathcal{S}^\mu} \sigma_i^z$. Finally, the Hamiltonian of the X-cube model can be written as

$$H_{\text{Xcube}} = -\sum_{\mathcal{C}} A_{\mathcal{C}} - \sum_{\mu} \sum_{\mathcal{S}^\mu} B_{\mathcal{S}^\mu} . \tag{3}$$

Just as the checkerboard model, the X-cube model is a stabilizer code with stabilizer generators $A_{\mathcal{C}}$ and $B_{\mathcal{S}^\mu}$. Hence, the model is solved exactly by $A_{\mathcal{C}} = B_{\mathcal{S}^\mu} = +1$ $\forall \mathcal{C}$ and $\mathcal{S}^\mu$. Again, one can express the ground state as

$$|\text{GS}_{\text{Xcube}}\rangle \propto \prod_{\mathcal{C}} \frac{(1 + A_{\mathcal{C}})}{2} |\uparrow \ldots \uparrow\rangle . \tag{4}$$

For periodic boundary conditions, the ground state is no longer unique and the degeneracy scales as $2^{6L-3}$ for an $L \times L \times L$ three-torus [2, 11].

The X-cube model hosts two different kinds of elementary excitations [2], which are illustrated in Figure 2b: First, immobile fractons that are created at the corners of $\sigma^z$-type membrane operators $M^z$. Second, so-called lineon excitations created at the ends and corners of $\sigma^x$-type string operators $W^x$. Remarkably, a lineon can move freely on a line along the direction $\mu$ indicated by the elongated dimension of the red ellipsoids in Figure 2b such that $B_{\mathcal{S}^\mu} = 1$ and $B_{\mathcal{S}^{\nu \neq \mu}} = -1$. Moreover, Figure 2b shows two pairs of logical operators that act on the ground state manifold (assuming periodic boundary conditions) corresponding to extended line $\Gamma_{1,2}^x$ and membrane $M_{1,2}^z$ operators winding around the three-torus.

## 2.3 Haah's code

For Haah's code, two separate qubits denoted by $\sigma$ and $\mu$ are placed on every site of a three-dimensional cubic lattice. Then, the Hamiltonian is defined as

$$H_{\text{Haah}} = -\sum_{\mathcal{C}} A_{\mathcal{C}} - \sum_{\mathcal{C}} B_{\mathcal{C}} , \tag{5}$$

$$A_{\mathcal{C}} := \mu_j^z \mu_k^z \sigma_l^z \mu_m^z \sigma_n^z \sigma_p^z \sigma_q^z \mu_q^z , \tag{6}$$

$$B_{\mathcal{C}} := \sigma_i^x \mu_i^x \mu_j^x \mu_k^x \sigma_l^x \mu_m^x \sigma_n^x \sigma_p^x , \tag{7}$$

which is illustrated in Figure 3. Here, both $\sigma_i^\alpha$ and $\mu_i^\alpha$ with $\alpha \in \{x, y, z\}$ denote the Pauli matrices on site $i$. The $A_{\mathcal{C}}$ and $B_{\mathcal{C}}$ are the commuting stabilizer generators of the stabilizer code. Consequently, the codespace of the system is determined by $A_{\mathcal{C}} = B_{\mathcal{C}} = +1$ $\forall \mathcal{C}$. Its ground state has the form

$$|\text{GS}_{\text{Haah}}\rangle \propto \prod_{\mathcal{C}} \frac{(1 + B_{\mathcal{C}})}{2} |\uparrow \ldots \uparrow\rangle_\sigma \otimes |\uparrow \ldots \uparrow\rangle_\mu . \tag{8}$$

Under periodic boundary conditions, its ground state degeneracy GSD strongly depends on the system size and is bounded by $2^2 \leq \text{GSD} \leq 2^{4L-2}$. Despite no closed-form formula being available for all system sizes, for any given $L$ the precise GSD can be computed using a polynomial ring formalism [52, 53]. Haah's code is a so-called type-II fracton model, i.e. all excitations are completely immobile. Fractons are created locally by the action of $\sigma^x$ ($\mu^x$) or $\sigma^z$ ($\mu^z$) operators, violating the $A_{\mathcal{C}}$ or $B_{\mathcal{C}}$ stabilizer generators, respectively. Note, any local operator must create at least four fractons at once. In Haah's code, there are no string-like operators that can individually move these fractons or bound states thereof. Instead, spatially separated excitations are created at the four corners of fractal operators instead of membrane-like operators [2, 52].

### 2.4 Perturbed fracton model

The form of the fracton codes permits various dynamical terms. Here we consider the simplest case in which the checkerboard model is perturbed by a uniform external magnetic field. The perturbed checkerboard model is given by

$$H = -\sum_{\mathcal{C}} A_{\mathcal{C}} - \sum_{\mathcal{C}} B_{\mathcal{C}} - \vec{h} \sum_i \vec{\sigma}_i \,, \tag{9}$$

where $\vec{\sigma}_i = (\sigma_i^x, \sigma_i^y, \sigma_i^z)$ denotes the Pauli vector at site $i$. More general terms such as bond or plaquette couplings are allowed but left for future work.

## 3 Neural quantum states

Consider an arbitrary pure quantum state $|\psi\rangle$ that is element of an $N$-qubit Hilbert space $\mathcal{H} = \otimes_i^N \mathcal{H}_i = \otimes_i^N \mathbb{C}_i^2$. The state can be expanded in the computational $z$-basis $\{ \, | \, \sigma_1...\sigma_N \rangle \}_{\sigma_i = \uparrow, \downarrow}$ (or any other basis of choice) as $|\psi\rangle = \sum_{\boldsymbol{\sigma}} \langle \boldsymbol{\sigma} | \psi \rangle |\boldsymbol{\sigma}\rangle = \sum_{\boldsymbol{\sigma}} \psi(\boldsymbol{\sigma}) |\boldsymbol{\sigma}\rangle$, where $\boldsymbol{\sigma}$ is a shorthand for $\sigma_1...\sigma_N$. There are in total $2^N$ complex coefficients $\psi(\boldsymbol{\sigma})$ that must be normalized $\sum_{\boldsymbol{\sigma}} |\psi(\boldsymbol{\sigma})|^2 = 1$. To compress the many-body wave function, $\psi(\boldsymbol{\sigma})$ is now expressed in terms of a neural network with parameters $\boldsymbol{\theta}$, which maps input spin configurations to complex-valued wave function amplitudes $\psi_{\boldsymbol{\theta}}(\boldsymbol{\sigma})$. This approach, first described by [31], mitigates the issues arising from the exponentially large Hilbert space by modelling the wave function in a much smaller parameter space. In particular, the parameterized wave functions are typically not normalized to avoid summation over said Hilbert space. Then, approximating the ground state with energy $E_0$ amounts to finding the optimal parameters that minimize the variational energy

$$E(\boldsymbol{\theta}) = \frac{\langle \psi_{\boldsymbol{\theta}} | H | \psi_{\boldsymbol{\theta}} \rangle}{\langle \psi_{\boldsymbol{\theta}} | \psi_{\boldsymbol{\theta}} \rangle} \geq E_0 \,. \tag{10}$$

This is done by estimating the gradient of $E(\boldsymbol{\theta})$ with Monte Carlo samples from the Born distribution $|\psi_{\boldsymbol{\theta}}|^2$ and updating the parameters into the direction of steepest descent using stochastic reconfiguration, first introduced by [54]. Repeating this parameter update iteratively then leads to a local, ideally global, minimum of the energy landscape $E(\boldsymbol{\theta})$. Hence,

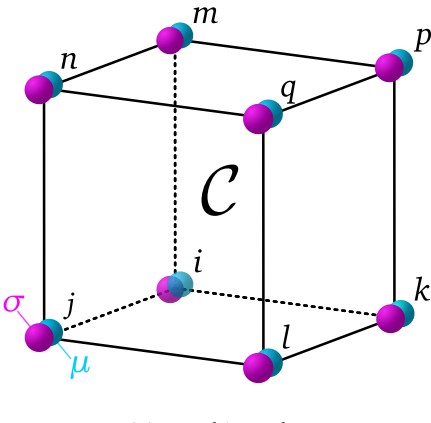

(a) Haah's code.

Figure 3: The positions of the qubits are indicated by the purple and cyan balls. Two separate qubits are placed on each lattice site and two stabilizer generators $A_{\mathcal{C}}$ and $B_{\mathcal{C}}$ are defined on each cube.

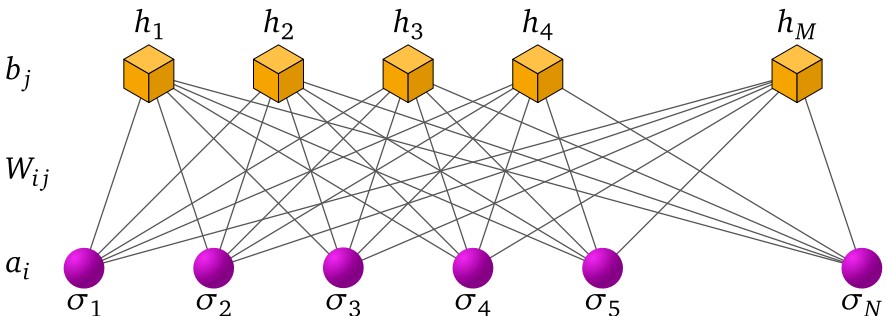

Figure 4: For an RBM, the $M$ hidden units (top) are connected to the $N$ visible units (bottom) through interactions described by the weight matrix $W_{ij}$. No connection between any two hidden or visible units is permitted. Moreover, both hidden and visible units are coupled to an inhomogeneous magnetic field via $b_j$ and $a_i$, respectively.

neural quantum states (NQS) are a subclass of variational quantum states characterized by parameterizing the wave function in terms of a neural network. Appendix B contains a more detailed description of the NQS optimization.

Although under which conditions a specific neural quantum state can parameterize a ground state of $H$ exactly remains a difficult question, it is in principle possible to improve the accuracy of the approximation arbitrarily by increasing the capacity of the neural network, which usually corresponds to increasing the number of variational parameters $\theta_i$. In the case of feedforward neural networks (FFNNs), for instance, the universal approximation theorem [47, 48] guarantees the approximation of arbitrary continuous (and even Lebesgue-integrable functions) when the width or depth of the network is increased appropriately.

We now introduce the neural network parametrizations used throughout this work. The restricted Boltzmann machine (RBM), a widely used NQS architecture [31, 35–37], is given by

$$
\begin{aligned}
\mathrm{RBM}_{\boldsymbol{\theta}}(\boldsymbol{\sigma}) &= \sum_{h_j = \pm 1} \exp\left( \sum_{i,j} W_{ji} h_j \sigma_i + \sum_{j=1}^{M} b_j h_j + \sum_{i=1}^{N} a_i \sigma_i \right) \\
&\propto \exp\left( \sum_{i=1}^{N} a_i \sigma_i \right) \prod_{j=1}^{M} \cosh\left( \sum_{i=1}^{N} W_{ji} \sigma_i + b_j \right).
\end{aligned}
\tag{11}
$$

Here, $\boldsymbol{\theta} = \{W, \boldsymbol{a}, \boldsymbol{b}\}$ collectively denotes the trainable parameters, the $M$ so-called hidden units $h_j \in \{\pm 1\}$ constitute the hidden layer of the RBM, and the $\sigma_i$ are termed visible units. Accordingly, we refer to the $a_i$ as visible biases and to the $b_j$ as hidden biases. Up to a normalization factor, this expression corresponds to the the canonical (or Boltzmann) distribution over the visible units after tracing out the hidden units, with the energy of the system given by $E(\boldsymbol{h}, \boldsymbol{\sigma})_{\mathrm{RBM}} = -\sum_{i,j} W_{ji} h_j \sigma_i - \sum_j b_j h_j - \sum_i a_i \sigma_i$. This architecture is illustrated in Figure 4.

In general, the parameters are complex numbers in order to model complex-valued wave function amplitudes. RBMs, just as FFNNs, are universal approximators [49]; by increasing the number of hidden units $M$, the accuracy of the approximation can be improved systematically.

Furthermore, we include translational symmetries into our Ansatz. The symmetrized wave function then satisfies $\langle g\boldsymbol{\sigma}|\psi_{\boldsymbol{\theta}}\rangle = \psi_{\boldsymbol{\theta}}(g\boldsymbol{\sigma}) = \psi_{\boldsymbol{\theta}}(\boldsymbol{\sigma}) \; \forall \boldsymbol{\sigma}$, where $g$ is an arbitrary translation on the corresponding lattice. This is achieved by sharing the weights in equation 11 over all translated versions of a given spin configuration, i.e. the orbit of the translation action on the spin configuration. See Appendix A for a detailed description of the symmetrized RBM architecture.

The main results of this work are produced using the correlation-enhanced RBM architecture (cRBM), introduced and demonstrated for the 2d toric code by [50]. To this end, we include correlators $C_i$, which are products of spins over specified index sets, into our neural network architecture. This is done by adding additional visible units representing the correlator values and connecting them to the hidden units with their own trainable parameters. The new energy functional then reads $E_{\text{cRBM}}(\boldsymbol{h}, \boldsymbol{\sigma}) = E(\boldsymbol{h}, \boldsymbol{\sigma})_{\text{RBM}} + \sum_i a_i^{\text{corr}} C_i + \sum_{i,j} W_{i,j}^{\text{corr}} C_i h_j$. This leads to the following expression for a cRBM with one type of correlators $C_i$:

$$\text{cRBM}_{\boldsymbol{\theta}}(\boldsymbol{\sigma}) = \exp\left(\sum_i a_i \sigma_i\right) \exp\left(\sum_i a_i^{\text{corr}} C_i\right) \prod_j \cosh\left(b_j + \sum_i W_{ji} \sigma_i + \sum_i W_{ji}^{\text{corr}} C_i\right). \quad (12)$$

In essence, one constructs a new feature vector out of the pure spin configuration, which is then fed into the neural network in its place.

The specific choice of correlators is guided by the available background knowledge about the system. For instance, the ground state of the unperturbed checkerboard model satisfies $B_{\mathcal{C}_i} = +1 \ \forall i$. Including these features explicitly into the variational Ansatz directly informs the network about the values of these stabilizer generators. Therefore, we make use of cube correlators $C_i = \prod_{\mathcal{C}_i} \sigma_i$, where the product is taken over the 8 spins at the corners of each cube $\mathcal{C}_i$. In addition to cube correlators, our final wave function Ansatz also includes bond and non-contractible loop correlators, as illustrated in Figure 5. Finally, the cRBM Ansatz is also symmetrized, see Appendix A for an explicit expression. To adapt the cRBM to the X-cube model, for instance, one should include the values of the $B_{\mathcal{S}^\mu}$ correlators. In this manner, the cRBM architecture can be tailored to any specific problem for which relevant correlations are known, resulting in improved performance over a regular RBM.

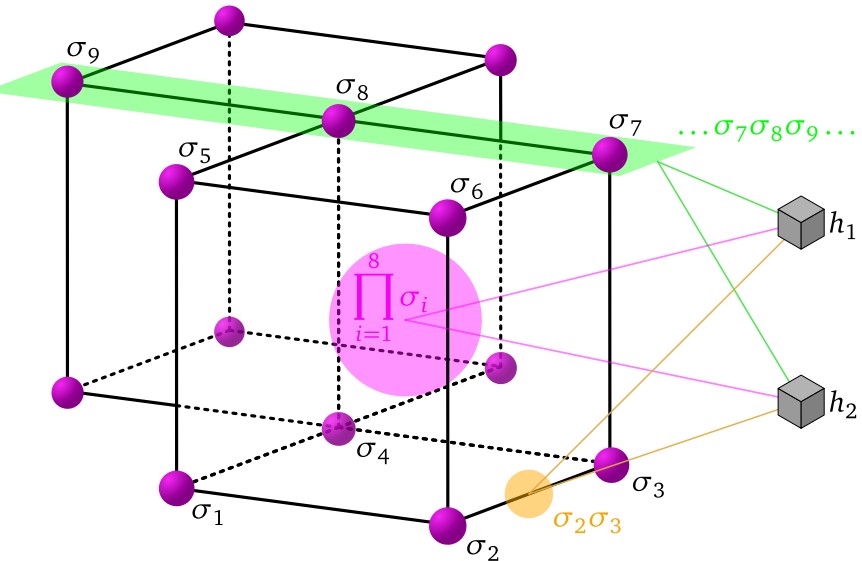

Figure 5: Illustrated is an example of the different correlators included in the cRBM architecture applied to the checkerboard model. A non-contractible loop correlator (green), cube correlator (purple) and bond operator (orange) are constructed from the input configuration and connected to the hidden units (grey).

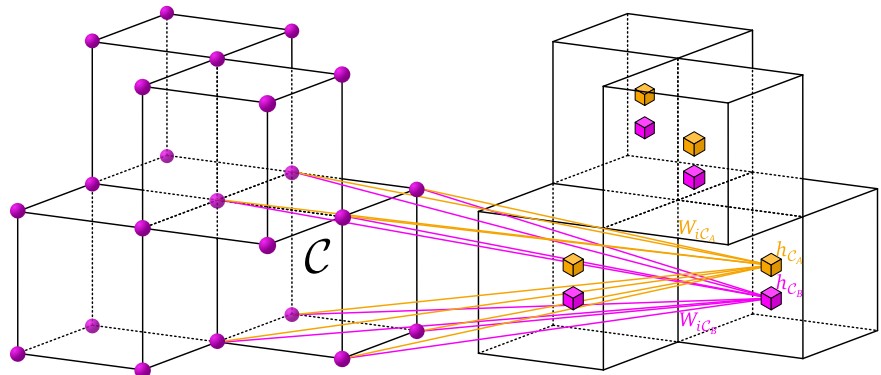

Figure 6: The hidden units (right) are connected locally to the cubes of the checkerboard model (left). We introduce two hidden units for each cube $\mathcal{C}$, here shown as orange and purple cubes.

# 4 Exact representations

In this section, we show how to construct exact and efficient RBM parametrizations for the checkerboard model, the X-cube model, and Haah's code. The key component is to impose sparsity on the weight matrix $W$ by connecting hidden units only locally to visible units. This results in an efficient parameter count that scales at most linearly in system size. Then, we construct their cRBM representations and show that including the right correlators can lead to significant simplifications. Throughout this work, we impose periodic boundary conditions (PBCs).

## 4.1 Checkerboard model

Introducing one hidden unit for each stabilizer generator and connecting them just locally to the 8 qubits on the vertices of the corresponding cubes, as illustrated in Figure 6, leads to

$$\psi_{\text{RBM}}(\boldsymbol{\sigma}) = \exp\left(\sum_i a_i \sigma_i\right) \prod_{\mathcal{C}_A} \cosh\left(b_{\mathcal{C}_A} + \sum_{i \in \mathcal{C}_A} \sigma_i W_{i\mathcal{C}_A}\right) \prod_{\mathcal{C}_B} \cosh\left(b_{\mathcal{C}_B} + \sum_{i \in \mathcal{C}_B} \sigma_i W_{i\mathcal{C}_B}\right). \quad (13)$$

Solving for the ground state amounts to finding the parameters such that $A_{\mathcal{C}} = B_{\mathcal{C}} = +1 \,\forall \mathcal{C}$, resulting in the conditions $\psi_{\text{RBM}}(\boldsymbol{\sigma}) = \prod_{k \in \mathcal{C}} \sigma_k \psi_{\text{RBM}}(\boldsymbol{\sigma})$ for any $B_{\mathcal{C}}$ and $\psi_{\text{RBM}}(\tilde{\boldsymbol{\sigma}}_{\mathcal{C}}) = \psi_{\text{RBM}}(\boldsymbol{\sigma})$ for any $A_{\mathcal{C}}$, where $\boldsymbol{\sigma}$ is arbitrary. Here, $\tilde{\boldsymbol{\sigma}}_{\mathcal{C}} = (\boldsymbol{\sigma}|\sigma_i \rightarrow -\sigma_i \,\forall i \in \mathcal{C})$ denotes the spin configuration after flipping all qubits belonging to $\mathcal{C}$.

We set the visible biases $a_i$ to zero and begin by solving the $B_{\mathcal{C}}$ stabilizer conditions, which are simpler than the $A_{\mathcal{C}}$ conditions because they are diagonal in the computational basis. This leads to

$$\cosh\left(b_{\mathcal{C}_B} + \sum_{i \in \mathcal{C}_B} \sigma_i W_{i\mathcal{C}_B}\right) = \prod_{i \in \mathcal{C}_B} \sigma_i \cosh\left(b_{\mathcal{C}_B} + \sum_{i \in \mathcal{C}_B} \sigma_i W_{i\mathcal{C}_B}\right) \,\forall \mathcal{C}_B, \quad (14)$$

after cancelling all other equal terms. Now, for any given cube $\mathcal{C}_B$ we observe that

$$\left(\sum_{k \in \mathcal{C}_B} \sigma_k, \prod_{k \in \mathcal{C}_B} \sigma_k\right) \in \{(0,1), (\pm2, -1), (\pm4, 1), (\pm6, -1), (\pm8, 1)\}. \quad (15)$$

By choosing $b_{\mathcal{C}_B} = 0$ and $W_{i\mathcal{C}_B} = W_{\mathcal{C}_B} = \mathrm{i}\frac{\pi}{4} \,\forall \mathcal{C}_B$, the conditions 14 are solved.

Now, in order to solve the $A_{\mathcal{C}}$ conditions, we consider flipping all qubits that belong to some cube $\mathcal{C}$. First, notice that the factor in Equation 13 corresponding to $\mathcal{C}$ itself is invariant under the flip and $\left(\sum_{k\in\mathcal{C}'}(\tilde{\sigma}_{\mathcal{C}})_k - \sum_{j\in\mathcal{C}'}\sigma_j\right)\in\{0,\pm4\}$ for any other cube $\mathcal{C}'\neq\mathcal{C}$. Hence, the $\mathcal{C}_B$-factors in Equation 13 stay invariant or gain a sign under such cube flips. To make the RBM Ansatz invariant, we simply choose the same set of parameters for the $\mathcal{C}_A$-terms as for the $\mathcal{C}_B$-terms: $b_{\mathcal{C}_A}=0$ and $W_{i\mathcal{C}_A}=W_{\mathcal{C}_A}=\mathrm{i}\frac{\pi}{4}$ $\forall\mathcal{C}_A$. Thus, our final expression for the ground state of the checkerboard model in terms of an RBM reads

$$\psi_{\text{RBM}}(\boldsymbol{\sigma})=\prod_{\mathcal{C}}\cos^2\left(\frac{\pi}{4}\sum_{i\in\mathcal{C}}\sigma_i\right).\tag{16}$$

We switch to the correlation-enhanced RBM (cRBM) architecture. In this setup, we only include cube correlators and set the weights coupling to the single visible units directly to zero. After connecting one hidden unit to the correlator feature of each cube and setting the visible biases to zero, the cRBM Ansatz can be expressed as

$$\psi_{\text{cRBM}}(\boldsymbol{\sigma})=\prod_{\mathcal{C}}\cosh\left(b_{\mathcal{C}}+W_{\mathcal{C}}\prod_{i\in\mathcal{C}}\sigma_i\right).\tag{17}$$

First, notice that the $A_{\mathcal{C}}$ stabilizer conditions are already satisfied. Flipping all qubits belonging to some cube $\mathcal{C}$ leaves this parametrization invariant as any $A_{\mathcal{C}}$ generator shares an even number of qubits with any $B_{\mathcal{C}}$ generator. The $B_{\mathcal{C}}$ conditions are easily solved by setting $b_{\mathcal{C}}=\mathrm{i}\frac{\pi}{4}$ and $W_{\mathcal{C}}=-\mathrm{i}\frac{\pi}{4}$ $\forall\mathcal{C}$. The final cRBM expression then reads

$$\psi_{\text{cRBM}}(\boldsymbol{\sigma})=\prod_{\mathcal{C}}\cos\left(\frac{\pi}{4}(1-\prod_{i\in\mathcal{C}}\sigma_i)\right).\tag{18}$$

The above RBM and cRBM representations contain $8N$ and $N$ non-zero parameters, respectively, hence scaling linearly in system size $N=L^3$.

## 4.2 X-cube model

Similar to the checkerboard model, one hidden unit is connected locally to every $\mathcal{S}^x,\mathcal{S}^y$ and $\mathcal{S}^z$ star corresponding to the $\sigma^z$-type stabilizer generators, as shown in Figure 7. We directly set all visible biases to zero to arrive at the following RBM Ansatz:

$$\psi_{\text{RBM}}(\boldsymbol{\sigma})=\prod_{\mu\in\{x,y,z\}}\prod_{\mathcal{S}^\mu}\cosh\left(b_{\mathcal{S}^\mu}+\sum_{i\in\mathcal{S}^\mu}\sigma_i W_{i\mathcal{S}^\mu}\right).\tag{19}$$

For any $\mathcal{S}^\mu$ with $\mu\in\{x,y,z\}$, we have

$$\left(\sum_{k\in\mathcal{S}^\mu}\sigma_k,\prod_{k\in\mathcal{S}^\mu}\sigma_k\right)\in\{(0,1),(\pm2,-1),(\pm4,1)\}.\tag{20}$$

Therefore, setting $b_{\mathcal{S}^\mu}=0$ and $W_{i\mathcal{S}^\mu}=W_{\mathcal{S}^\mu}=\mathrm{i}\frac{\pi}{4}$ $\forall\mathcal{S}^\mu$ solves the $B_{\mathcal{S}^\mu}$ stabilizer conditions $\psi_{\text{RBM}}(\boldsymbol{\sigma})=\prod_{k\in\mathcal{S}^\mu}\sigma_k\psi_{\text{RBM}}(\boldsymbol{\sigma})$ for any $\mathcal{S}^\mu$. This leads to the following (and indeed final) form of the exact RBM parametrization

$$\psi_{\text{RBM}}(\boldsymbol{\sigma})=\prod_{\mu\in\{x,y,z\}}\prod_{\mathcal{S}^\mu}\cos\left(\frac{\pi}{4}\sum_{i\in\mathcal{S}^\mu}\sigma_i\right).\tag{21}$$

Regarding the $A_{\mathcal{C}}$ conditions, consider flipping all 12 qubits belonging to some cube $\mathcal{C}$ and focus on any corner of $\mathcal{C}$ with the three qubits associated to that corner, see Figure 7. At this

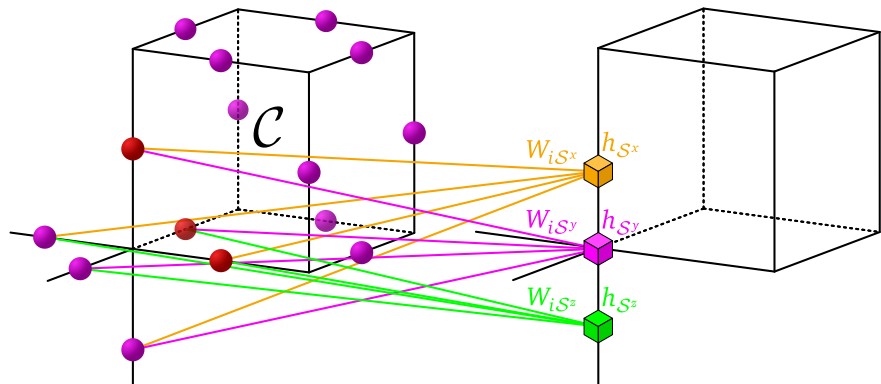

Figure 7: The hidden units (right) are connected locally to the three different stars $\mathcal{S}^x, \mathcal{S}^y$ and $\mathcal{S}^z$ at each corner in the X-cube model (left). The three qubits in $\mathcal{C}$ associated with the corner in the bottom left are colored red.

vertex, there is exactly one $\mathcal{S}^x, \mathcal{S}^y$ and $\mathcal{S}^z$ affected by the flip. If all three corner qubits have the same value, each factor in Equation 21 corresponding to the three $\mathcal{S}^\mu$ gains a sign. If one qubit is different than the other two, only the one $\mathcal{S}^\mu$-factor containing the two equal qubits gains a sign. Hence, the wave function 21 acquires one sign for each corner of the flipped cube, so the $A_\mathcal{C}$ stabilizer conditions are directly satisfied.

To find the expression of the X-cube ground state in terms of a cRBM, we introduce star correlators for all $\mathcal{S}^\mu$ and connect one hidden unit to each of them. After setting all visible biases to zero, we get

$$\psi_{\text{cRBM}}(\boldsymbol{\sigma}) = \prod_{\mu \in \{x,y,z\}} \prod_{\mathcal{S}^\mu} \cosh\left( b_{\mathcal{S}^\mu} + W_{\mathcal{S}^\mu} \prod_{i \in \mathcal{S}^\mu} \sigma_i \right). \tag{22}$$

Conveniently, the $A_\mathcal{C}$ stabilizer conditions are directly satisfied as each cube flip changes the sign of either none or two qubits belonging to any $\mathcal{S}^\mu$. To solve the $B_{\mathcal{S}^\mu}$ stabilizer conditions, we simply choose $b_{\mathcal{S}^\mu} = i\frac{\pi}{4}$ and $W_{\mathcal{S}^\mu} = -i\frac{\pi}{4}$ $\forall \mathcal{S}^\mu$ in analogy to the checkerboard model. Thus, the cRBM parametrization of the X-cube ground state reads

$$\psi_{\text{cRBM}}(\boldsymbol{\sigma}) = \prod_{\mu \in \{x,y,z\}} \prod_{\mathcal{S}^\mu} \cos\left( \frac{\pi}{4}(1 - \prod_{i \in \mathcal{S}^\mu} \sigma_i) \right). \tag{23}$$

Again, the cRBM parametrization contains less non-zero parameters, i.e. $6N$, compared to the RBM Ansatz with $12N$ parameters. For completeness, we point to [55] for an alternative parametrization of the X-cube ground state. At this point, the general scheme for constructing an exact and efficient cRBM parametrization for any CSS code should become clear: Introduce a correlator feature for every $\sigma^z$-type stabilizer generator $T_i^z$ and locally connect a hidden unit to each of them with corresponding weights $b_i = i\frac{\pi}{4}$, $W_i = -i\frac{\pi}{4}$ and visible biases set to zero. For a stabilizer code, all $\sigma^x$-type stabilizer generators $T_i^x$ commute with any $T_j^z$, so they must share an even number of qubits. Hence, the $T_i^x$ generator conditions are directly satisfied.

## 4.3 Haah's code

To arrive at an RBM representation of Haah's code, we introduce two hidden units for each cube $\mathcal{C}$ and connect them locally to the sites on which the $\sigma^z$-type stabilizer generator $A_\mathcal{C}$ acts non-trivially, as is illustrated in Figure 8. After setting the visible biases to zero and sharing

the weights over the corresponding sites within each cube, we arrive at the following Ansatz:

$$\psi_{\text{RBM}}(\boldsymbol{\sigma},\boldsymbol{\mu}) = \prod_{\mathcal{C}} \cosh\left(b_{\mathcal{C}} + W_{\mathcal{C}}(\mu_j + \mu_k + \sigma_l + \mu_m + \sigma_n + \sigma_p + \sigma_q + \mu_q)\right)$$
$$\times \cosh\left(b'_{\mathcal{C}} + W'_{\mathcal{C}}(\mu_j + \mu_k + \sigma_l + \mu_m + \sigma_n + \sigma_p + \sigma_q + \mu_q)\right). \tag{24}$$

For brevity, the indexing of the sites is always relative to the cube index of the corresponding cosh-factor; for instance, we write $\mu_j$ instead of $\mu_{\mathcal{C},j}$.

The exact parameters are found in complete analogy to the checkerboard model: To satisfy the $A_{\mathcal{C}}$ stabilizer conditions, we choose $b_{\mathcal{C}} = 0$ and $W_{\mathcal{C}} = \mathrm{i}\frac{\pi}{4}$ $\forall \mathcal{C}$. Next, we observe that any $B_{\mathcal{C}}$ operator shares 6 qubits with $A_{\mathcal{C}}$ and either two or no qubits with any $A_{\mathcal{C}'}$, where $\mathcal{C}'$ is any cube adjacent to $\mathcal{C}$. It follows from some quick combinatorial considerations that any cosh-factor corresponding to $b_{\mathcal{C}}$ and $W_{\mathcal{C}}$ stays either invariant or changes sign by the action of $B_{\mathcal{C}}$. Hence, the $B_{\mathcal{C}}$ stabilizer conditions are solved simply by setting $b_{\mathcal{C}'} = b_{\mathcal{C}}$ and $W_{\mathcal{C}'} = W_{\mathcal{C}}$. Then, the exact RBM representation of Haah's code reads

$$\psi_{\text{RBM}}(\boldsymbol{\sigma},\boldsymbol{\mu}) = \prod_{\mathcal{C}} \cos^2\left(\frac{\pi}{4}(\mu_j + \mu_k + \sigma_l + \mu_m + \sigma_n + \sigma_p + \sigma_q + \mu_q)\right). \tag{25}$$

Finally, we can write down the exact cRBM representation in a straightforward way: We introduce one hidden unit for each $A_{\mathcal{C}}$ stabilizer generator, connect them to the corresponding correlators, and choose $b_{\mathcal{C}} = \mathrm{i}\frac{\pi}{4}$ and $W_{\mathcal{C}} = -\mathrm{i}\frac{\pi}{4}$ $\forall \mathcal{C}$. This results in

$$\psi_{\text{cRBM}}(\boldsymbol{\sigma}) = \prod_{\mathcal{C}} \cos\left(\frac{\pi}{4}(1 - \mu_j \mu_k \sigma_l \mu_m \sigma_n \sigma_p \sigma_q \mu_q)\right). \tag{26}$$

This Ansatz contains $N$ non-zero parameters, much less than the RBM parametrization with $8N$ parameters.

The linear scaling in system size of all derived exact parametrizations can be made constant by further imposing translational symmetries. For instance, a translation invariant RBM architecture with two hidden units is capable of representing the ground state of the unperturbed checkerboard model exactly, as shown in Figure 5 and Appendix A. Such knowledge of exact representations can guide the architecture design for perturbed fracton models where exact solutions become unavailable.

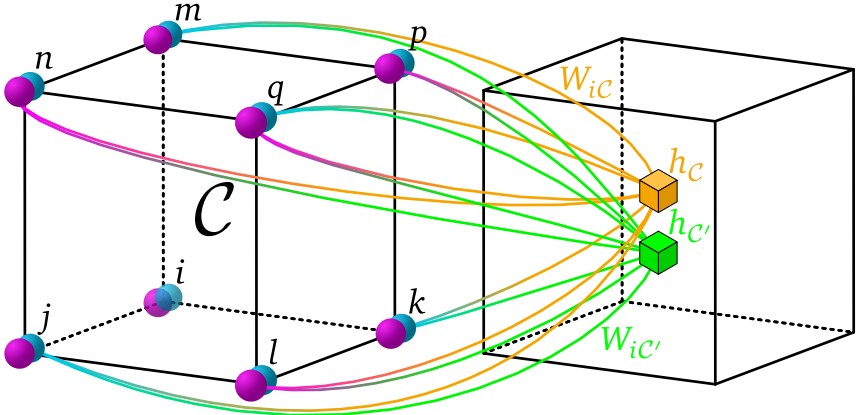

Figure 8: The hidden units (right) are connected locally to the cubes of Haah's code (left). We introduce two hidden units for each cube $\mathcal{C}$, here shown as orange and purple cubes.

# 5 Simulation overview

We utilize Netket [56] as the infrastructure of our code and use JAX [57] to exploit GPU acceleration. A translation-invariant cRBM Ansatz is employed and detailed in Appendix A. The simulations are performed on A100 and V100 GPUs. We typically use $2^{12}$–$2^{14}$ samples per epoch, distributed on $2^{10}$ individual Markov chains, to train the cRBM. The training time of one cRBM is about 6 A100-hours for the largest system size $L = 8$ (512 qubits). $24 \times 2^{14}$ reasonably-thermalized samples are considered to compute expectation values of physical observables.

The Netket library provides a flexible and convenient framework to implement different Hamiltonians and operators through its local operator interface. Nonetheless, the large-weight stabilizer generators of the checkerboard model allow for significant performance improvements. We provide a custom implementation of the computation of connected states and corresponding matrix elements in a jax-compatible way, which reduces the computation time by orders of magnitudes as compared in Appendix C with Netket's built-in implementation.

Moreover, we introduce a transfer learning protocol that carries over samples and parameters of a trained NQS to the optimization for the next physical parameter, namely, a nearby value of the magnetic field $\vec{h}$, see Appendix B. By comparing the results for transfer learning along increasing and decreasing fields, the method detects hysteresis effects in the checkerboard model and provides insights into the nature of the underlying phase transitions. We verified this method on the X-cube model, for which existing quantum Monte Carlo and high-order series expansion results predict a strong first-order transition, see Appendix E.2.

Sampling schemes like the Metropolis-Hasting algorithm can suffer from many issues such as long auto-correlation times, for instance. We make use of a custom update rule including, for example, cube flips (see Appendix B), perform sampling on $O(1000)$ chains in parallel, and carefully analyze the convergence of our simulations by monitoring behaviors of the acceptance ratio of our update rule, the split-$\hat{R}$ value [58], and the V-score [59] in Appendix D.

We refer to the Appendices for details of the complexity and the optimization algorithms and Repository [51] for the code implementation.

# 6 Benchmarking on small lattices

We benchmark the performance of the cRBM on a checkerboard model of size $4 \times 2 \times 2$ with PBCs subject to uniform magnetic fields along different directions. Such a system size remains exactly diagonalizable, hence no sampling is required and we can compare with the exact results.

First, in Figure 9 we compare the performance of several common NQS architectures on the pure checkerboard model in the absence of magnetic fields and for different combinations of hyperparameters, namely the standard deviation of the parameter initialization and the learning rate. We make use of the following architectures: an FFNN with two hidden layers containing 8 hidden units each, a regular RBM with 16 and 8 hidden units, a translation-invariant RBM with 8 and 4 hidden units, a symmetric FFNN that first constructs 4 translation-invariant features (as in a symmetric RBM) which are then processed by another layer consisting of 4 hidden units before being accumulated, a Jastrow Ansatz, and a translation-invariant cRBM with $\alpha = 1/4$ containing only additional cube correlators. The Jastrow wave function is modeled as $\psi_{\text{Jastrow}}(\boldsymbol{\sigma}) = \exp\left(\sum_{i \neq j} \sigma_i W_{ij} \sigma_j\right)$ with a symmetric matrix $W$ that is treated as lower triangular. All types of RBM architectures except RBM(8) contain enough hidden units to learn the exact representations derived in Section 4.

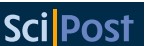

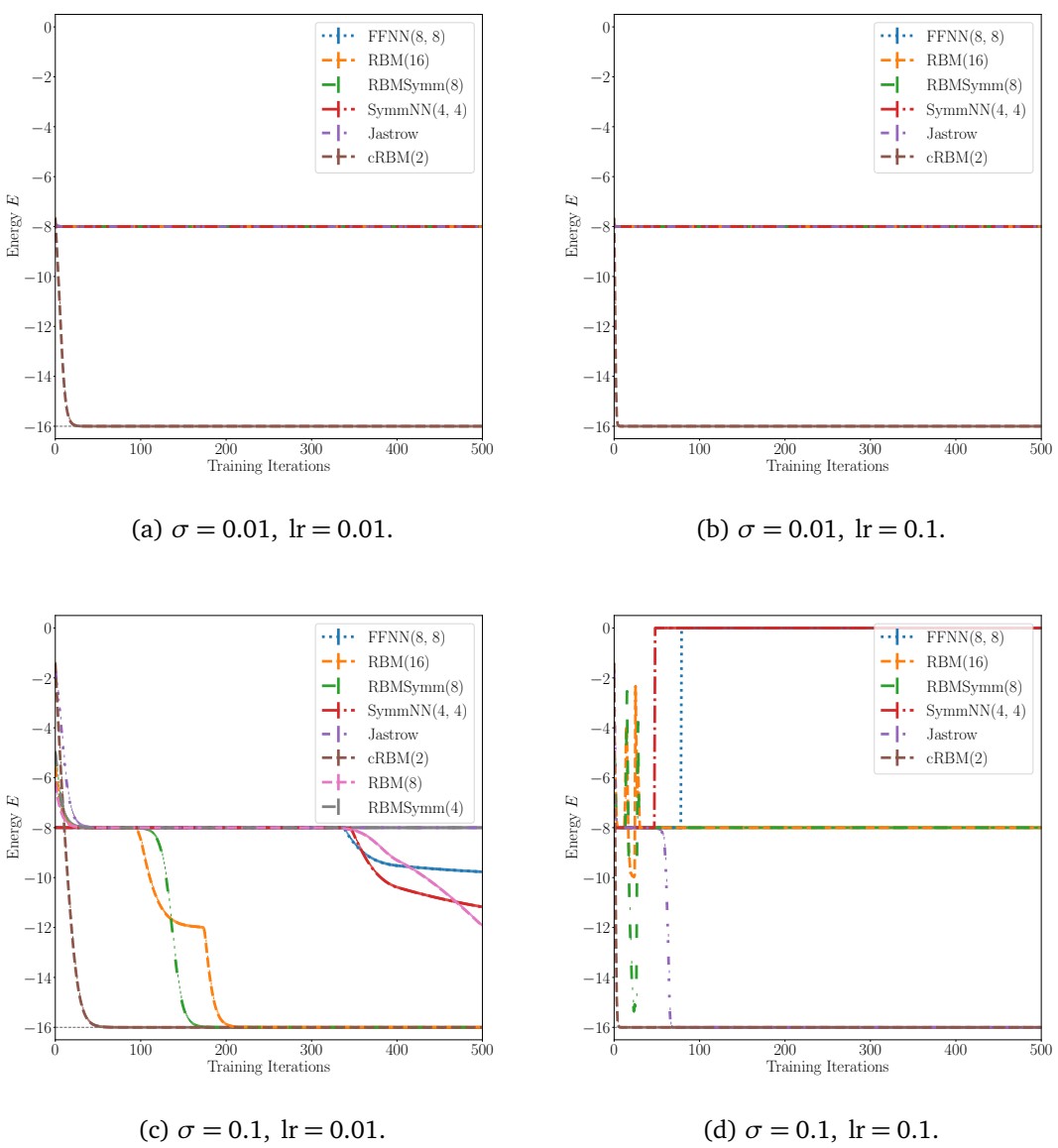

Figure 9: Comparison of different model architectures for learning the ground state of the pure Checkerboard model on a $4 \times 2 \times 2$ lattice. The complex parameters are sampled independently from a centered complex Gaussian with given standard deviation. The number in parentheses corresponds to the number of hidden units. The parameter counts for the different models are 208 for the FFNN, 288/152 for the RBM(16)/(8), 137/69 for the symmetric RBM(8)/(4), 88 for the symmetric NN, 120 for the Jastrow and 52 for the cRBM architecture. The ground state energy is indicated by the dashed grey line.

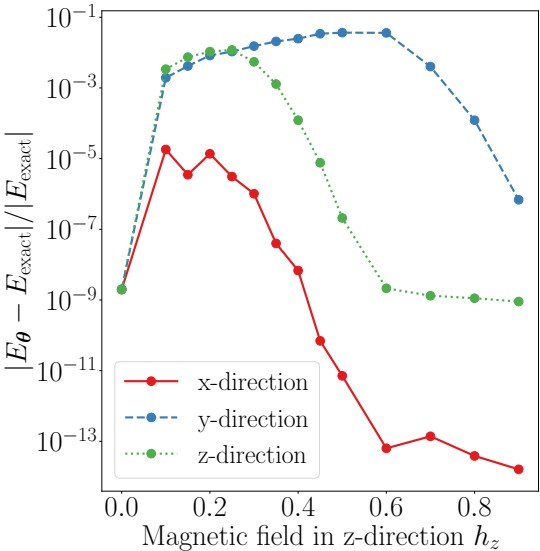

Figure 10: Relative energy error of the trained cRBM for different field strengths and directions, working in the computational $z$-basis. For every field strength separately, the initial parameters of the model are sampled independently from a Gaussian distribution $\mathcal{N}(0, \sigma^2)$ with small standard deviation $\sigma \sim 10^{-2}$.

The initial complex-valued parameters are sampled independently from a complex Gaussian distribution $\mathcal{N}(0, \sigma^2)$ with zero mean and standard deviation $\sigma$, a common practice in machine learning. For sufficiently small $\sigma$, this implies that the initial state described by any network with a final cosh-activation function approximately corresponds to the equal-weighted superposition of all configurations $|\psi_{\theta \approx 0}\rangle_z \approx 2^{-\frac{N}{2}} \otimes_i (|{+}z\rangle_i + |{-}z\rangle_i) = \otimes_i |{+}x\rangle_i$. Consequently, all $A_{\mathcal{C}}$ stabilizer conditions are approximately satisfied, resulting in an initial energy of $E(\theta_0) \approx E_0/2$. This state, however, appears to be a strong local minimum for all but the cRBM architecture, as can be seen in Figure 9a. The challenge of transitioning from an equal weighted superposition of all states to a fracton soup, the equal weighted superposition of states which also satisfy the $B_{\mathcal{C}}$ stabilizer conditions, is lifted for the cRBM by simplifying its loss landscape through the inclusion of the cube correlators.

By increasing the variance of the parameter initialization as in Figure 9c, it is possible to move away from said local minimum, thereby allowing the (symmetric) RBM architectures with $\alpha = 1$ to reach the ground state within the given number of training iterations. A larger variation in the initial parameters can result in unstable training and may lead to local minima later in the optimization, especially for larger learning rates as in Figure 9d. Also, we refer to Figure 20 in Appendix E.1 which displays the training curves for uniform parameter initialization. Figure 9c additionally shows training curves for the (symmetric) RBM architectures with only half as many hidden units. Clearly, the reduced $\alpha$ value results in non-convergence of these RBM architectures within the given number of training steps, although both models still contain more parameters than the cRBM. In conclusion, while other network architectures are able to approximate the ground state successfully in some scenarios, the cRBM architecture displays significant advantage over the others: it requires the smallest number of training iterations to learn the ground state, it exhibits strong robustness to different hyperparameter choices, it typically reaches the lowest energy variance by the end of training, and it is the most parameter efficient parametrization of the considered candidates. This highlights the significant advantage achieved by incorporating domain knowledge.

Next, relative energy errors of the trained cRBM for the checkerboard model perturbed with different fields (Eq. 9) are shown in Figure 10. Implementation details and hyperparameter choices are contained in Appendix C. We observe a significant difference in accuracy between magnetic fields applied along the $x$- and $z$-directions, although the only difference between these two field configurations is a rotation from the $x$- to the $z$-basis. Hence, this discrepancy must be due to the choice of basis made when expanding the wave function, the only step where the $x$- and $z$-directions are treated differently. Of course, the choice of basis inherently determines the form of the loss landscape $E(\boldsymbol{\theta})$, and hence the difficulty of the optimization problem. In particular, however, it determines the initial state in the following sense: Here, the initial parameters were sampled independently from a Gaussian distribution $\mathcal{N}(0, \sigma^2)$ with zero mean and small standard deviation $\sigma \sim 10^{-2}$. In the $z$-basis, this implies that the initial state described by an RBM or cRBM approximately corresponds to the equal-weighted superposition of all configurations $|\psi_{\boldsymbol{\theta}\approx0}\rangle_z \approx 2^{-\frac{N}{2}} \otimes_i (|+z\rangle_i + |-z\rangle_i) = \otimes_i |+x\rangle_i$. On the other hand, when working in the $x$-basis this parameter initialization results in an initial state that is close to the $z$-polarized state $|\psi_{\boldsymbol{\theta}\approx0}\rangle_x \approx 2^{-\frac{N}{2}} \otimes_i (|+x\rangle_i + |-x\rangle_i) = \otimes_i |+z\rangle_i$. Together with the results in Figure 10, this suggests that the performance for some applied magnetic field might be improved by initializing the NQS close to the corresponding polarized state.

With that motivation in mind, we introduce a simple transfer-learning protocol: First, the NQS is optimized in the presence of strong magnetic fields, which can be done for all field directions with high accuracy (see Figure 10). Then, the NQS is trained for the next highest field strength with the previously optimized NQS as the initial state. We achieve this by taking the learned parameters and, whenever full summation over the Hilbert space is not possible due to larger system sizes, also the final states of the Markov chains as the starting point for the NQS optimization with the new magnetic field. This process is repeated until some smallest field strength, usually zero, is reached. Hence, we not only initialize the NQS close to the polarized state but we also recycle learned features by transferring the network parameters. We would like to point out that this optimization scheme can also be understood as an implementation of variational neural annealing (VNA) [60]. A similar protocol has also been utilized in [61]. See Appendix B for additional details.

The relative energy error of the cRBM applied to the $4 \times 2 \times 2$ checkerboard model for different field directions and transfer-learning protocols is shown in Figure 11. In addition to transfer-learning from strong to weak fields, we compare against the same protocol for increasing field strengths. We observe significant performance improvements for all field directions if we initialize in the polarized phase and transfer to weaker fields (right-left sweep), while initializing in the absence of magnetic fields and transferring to stronger fields (left-right sweep) leads only to minor improvements except for increasingly strong $y$-fields. This demonstrates that the accuracy improvement is not solely caused by reusing the network parameters and the initial state of the transfer-learning sequence is indeed crucial.

# 7 Field-driven phase transitions

We now move on to larger systems of size $L \times L \times L$, $L \in \{4, 6, 8\}$, corresponding to 64, 216 and 512 qubits, respectively. The low parameter count of the cRBM, i.e. 5195 for $L = 8$, makes these computations feasible even with limited computational resources. The observables and gradients are estimated with samples obtained from the Metropolis algorithm; in addition to single spin flips, our update rule includes cube and non-contractible loop flips, see Appendix B and C for further technical details.

First, we inspect the results for the perturbed checkerboard model Eq. 9 under uniform $h_x$ and $h_z$ fields on an $L = 4$ lattice. The performance issue for $h_z$-fields is in principle mitigated

by transfer learning from strong to weak fields. This is indeed the case as indicated by the comparisons of energies and magnetizations along the two field directions in Figure 12. Still, slightly lower energies are achieved for $h_x$-fields in the region $0.37 \lesssim h \lesssim 0.45$, which is consistent with the previous benchmark results in Fig. 11. Note, as the checkerboard model is self-dual, the two field directions are physically equivalent upon changing basis. Hence, from now on we focus on $h_x$-fields.

Moreover, although the benchmark results in Fig. 11 suggest better performance when sweeping from strong to weak fields (right-left), we also include sweeps from weak to strong fields (left-right) for the following reason: Left-right sweeps, in particular along the $x$-direction, still result in a reasonably high energy accuracy, and comparing the two sweep directions will help uncover a hysteresis.

The kick in energy and jump in magnetization in Figure 12 suggest a first-order phase transition for both field directions already at $L = 4$. Nonetheless, to confirm its first-order nature, one has to verify that these signatures get sharper with increasing system sizes.

As shown in Figure 13 for lattice sizes $L = 4, 6, 8$, the energy and magnetization hystereses become more profound, as revealed by comparing the left-right and right-left transfer learning results. When sweeping from right to left, the NQS settles in a meta-stable phase resulting in higher energy for fields slightly weaker than the critical value $h_{\text{crit}}$. When the field strength is decreased further, the state becomes unstable and collapses into a new state with lower energy. This sudden jump coincides with a discontinuous decrease of the magnetization. For left-right sweeps we observe similar behavior, but for system sizes $L > 4$ the NQS remains in the meta-stable phase for all observed field strengths. This confirms that the perturbed checkerboard model indeed experiences a strong first-order phase transition. We estimate a critical field $h_{\text{crit}} \approx 0.44(1)$ from the intersection points of energy curves.

Note that we begin the left-right transfer-learning sequence for $L = 6, 8$ system sizes with $h = 0.1$ instead of the zero field limit. This is because we observe that pre-training the NQS in the absence of magnetic fields typically led to no training progress or diverging gradients after transferring to finite magnetic fields. Some literature, such as Ref. [50], suggests adding noise to the parameters when transferring to non-zero magnetic fields to mitigate this issue; however, in our case we could not find an improvement compared to directly optimizing the NQS from scratch.

Consistently, as the system is trapped in different (meta-)stable configurations near the first-order transition, the split-$\hat{R}$ diagnostic and energy variance also display a significant increase in magnitude; these and other MC convergence measures are included in App. D. This is simply a consequence of slowly-mixing Markov chains caused by the two coexisting phases which is, however, mitigated well by sampling from $O(1000)$ chains in parallel.

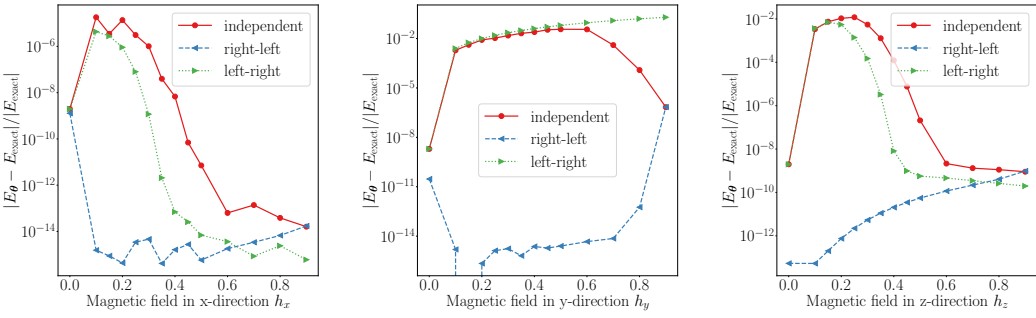

Figure 11: Relative energy error of the optimized cRBM for different transfer learning protocols and external magnetic fields on the $4 \times 2 \times 2$ checkerboard model.

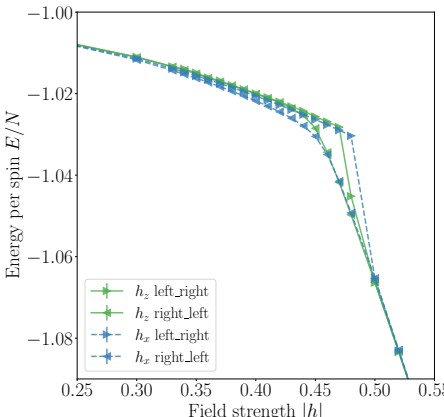
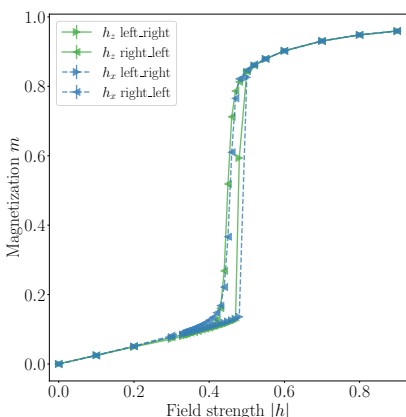

Figure 12: Comparison of the energy per spin and magnetization obtained from the symmetric cRBM architecture trained on the checkerboard model for $L = 4$ and magnetic fields along the $x$- and $z$-directions.

For a deeper understanding of the phase transition, we show the relative energy contributions of different parts of the Hamiltonian in Figure 14. The perturbation commutes with the $\sigma^{(x)}$-type stabilizer generators $A_{\mathcal{C}}$, leaving their eigenvalues approximately invariant in the fracton phase. Clearly, there is a transition into an $x$-polarized phase, which is essentially dominated by the contributions of the magnetic field and $A_{\mathcal{C}}$ stabilizer generators. Intuitively, this behaviour can be understood by condensing fracton excitations: In a perturbative picture, small magnetic fields create isolated fracton excitations. This violates the corresponding $B_{\mathcal{C}}$ stabilizer generators, thereby slightly reducing their contribution to the variational energy. Creating enough excitations through an increasingly strong magnetic field then effectively lifts their mobility restrictions. This results in a significant violation of the $B_{\mathcal{C}}$ stabilizer generators and leads to a vanishing energy cost for creating further excitations.

Here we do not use the Wilson loop, i.e. a non-contractible loop as in Figure 5, as an effective order parameter because of the limited system sizes. Although 512 qubits is a formidable number, a linear lattice extend of $L = 8$ may be too small to faithfully determine a perimeter law or an area law, where the loop correlators decay exponentially in both cases [62]. Nevertheless, we expect fractons to remain (partially) deconfined at small fields and become confined in the polarized phase, as in the case of the type-I X-cube model [63].

The situation for finite magnetic fields in $y$-direction is more complicated: First of all, the Hamiltonian becomes non-stoquastic. Although we employ complex-valued parameters, which in principle enable the network to express complex-valued wave function amplitudes, it is known that learning the structure of the phase factors is a difficult task for many neural network architectures [64–66]. Indeed, when attempting a right-left sweep for system size $4 \times 4 \times 4$, we observe a very high variance in the same order of magnitude as the variational energy for strong $y$-fields $\approx 0.9$. This quickly leads to diverging gradients if the optimization is not regularized accordingly. We find that the choice of solver for stochastic reconfiguration, iterative conjugate gradient or direct pseudo-inverse, has no significant impact on this issue. Moreover, all sampling diagnostics indicate high-quality Markov chains with $\hat{R} \lesssim 1.01$, acceptance rates of at least 0.4, and auto-correlation times $\tau \lesssim 0.1$. Hence, it appears that the network architecture itself has difficulty learning the structure of the phase factors even for the simple $h_y$-polarized phase. This observation is in accordance with [64], who found that complex RBMs have difficulty representing the ground states of Hamiltonians that cannot be transformed into a stoquastic form with local Pauli and phase-shift transformations.

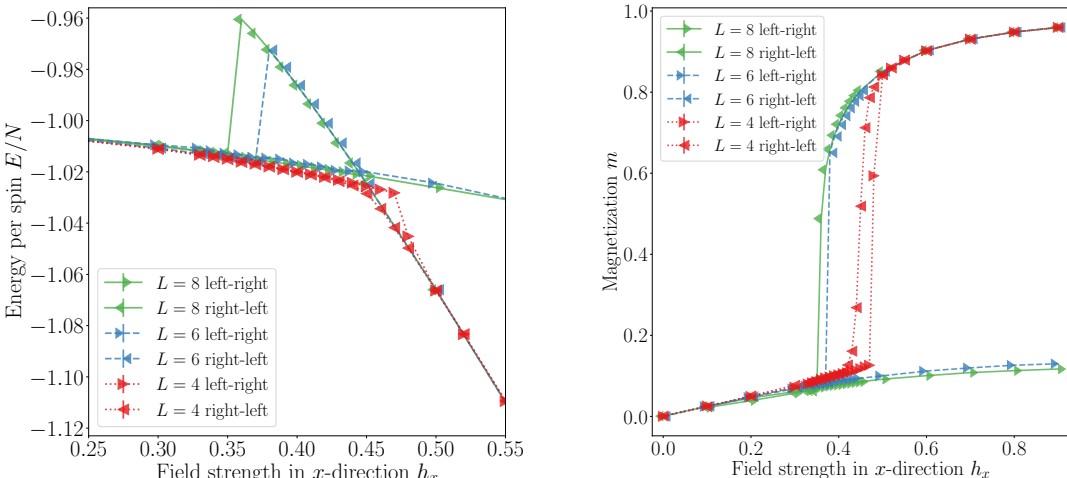

Figure 13: Comparison of the energy per spin and magnetization obtained from the symmetric cRBM architecture trained on the checkerboard model for different system sizes ($L = 4$, $L = 6$, $L = 8$) and sweep directions.

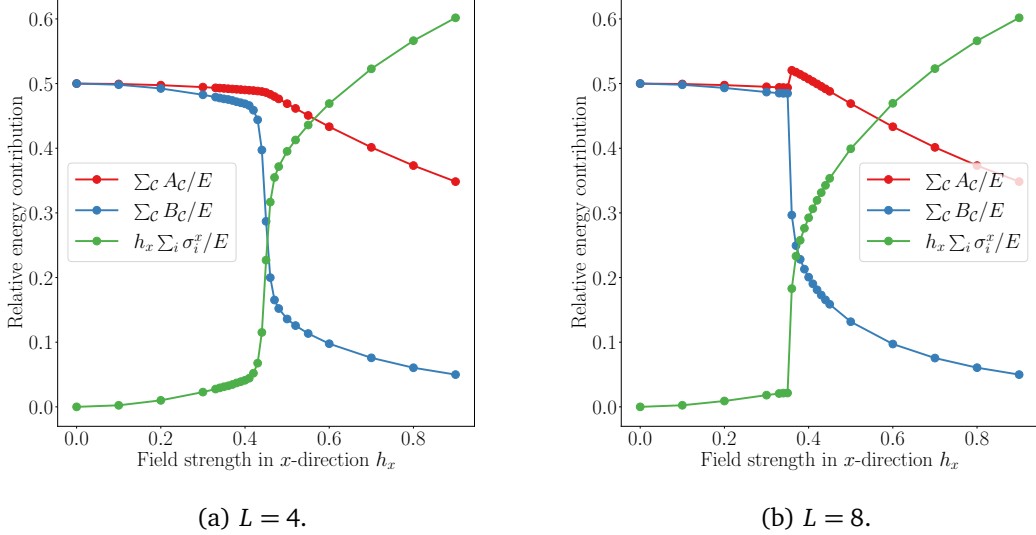

(a) $L = 4$.

(b) $L = 8$.

Figure 14: Relative contributions of different parts of the Hamiltonian to the ground state energy of the $L = 4$ and $L = 8$ Checkerboard model subject to a magnetic field in $x$-direction obtained from a right-left sweep.

This problem may be amended, as proposed in [65, 66], by splitting the network into two separate components, one of which encodes the phase and the other one the modulus of the wave function amplitudes. Nevertheless, we leave the systematic study of $h_y$-fields to future endeavors.

## 8 Conclusion

In this work, we explored the utility of neural quantum states in studying three-dimensional topological fracton models and performed extensive simulations for the perturbed checkerboard model for up to 512 qubits. We finalize our discussions by revisiting the two objectives of this paper.

On the side of fracton physics, our work disclosed that the checkerboard model experiences a strong first-order phase transition with a large hysteresis when subjected to uniform magnetic fields. This finding leads to an interesting observation: All three prototypical 3D lattice fracton models, including the type-I X-cube [44] and checkerboard models and the type-II Haah's code [46], show strong first-order phase transitions against the simplest quantum fluctuations. The same is true for their dual classical spin models, which correspond to the ungauged version of fracton models [2] and experience discontinuous thermal phase transitions [52, 67]. This may imply a ubiquitous relation between subsystem symmetries and first-order transitions, and mark intrinsic challenges to the development of field theories and mean-field analyses.

On the side of simulation techniques, we show that NQS methods are promising for studying complicated three-dimensional problems. They are able to approximate the ground states of long-range entangled 3D systems beyond the exactly solvable limits for considerably large lattice sizes to high accuracy. Nevertheless, physical insights play a crucial guide for the construction of efficient NQS representations. In the current example, implementing well-chosen correlator features using the cRBM architecture has led to significant performance improvements. Hence, our work highlights the importance of physics-oriented NQS design.

Still, some challenges remain: Our approach suffers from increased variance in the critical region, which we believe is mainly due to sampling issues caused by the phase separation near the strong first-order transition. In general, we believe it is necessary to explore more complex network architectures that better exploit the structure and symmetries of the problem of interest to further reduce the variance and arrive at more accurate results in the critical regime. In addition, the performance issues for the non-stoquastic Hamiltonians require further investigation. While the benchmarking results on the small system suggest high accuracy is possible in the presence of $\sigma_y$ coupling, we found that scaling up the NQS for larger system sizes suffers from training instabilities / non-convergence.

NQS constitutes a relatively young field of computational physics that often finds itself in competition with tensor networks and quantum Monte Carlo methods. Although NQS achieve state-of-the-art ground state energies for many different benchmark problems [59, 68–70], we want to establish NQS as viable tools for unsolved problems which are not readily accessible by other commonly employed numerical methods. With this work, we made a step towards achieving this goal by demonstrating that even a simple NQS architecture and carefully implemented domain knowledge can lead to new insights into complex three-dimensional systems.

## Acknowledgments

We thank Simon Linsel and Miguel A. Martin-Delgado for insightful discussions. Moreover, we want to thank Agnes Valenti for valuable input about the training behaviour of the cRBM on the toric code and Filippo Vicentini for technical support regarding NetKet.

**Funding information** M.M. K.L., and L.P. acknowledge support from FP7/ERC Consolidator Grant QSIMCORR, No. 771891. This work is part of the funding initiative Munich Quantum Valley, which is supported by the Bavarian state government with funds from the Hightech Agenda Bayern Plus, and by the Deutsche Forschungsgemeinschaft (DFG, German Research Foundation) under Germany's Excellence Strategy - EXC-2111 - 390814868. K.L. acknowledges support from the New Cornerstone Science Foundation through the XPLORER PRIZE, Anhui Initiative in Quantum Information Technologies, Shanghai Municipal Science and Technology Major Project (Grant No. 2019SHZDZX01), and the National Natural Science Foundation of China (Grant No. 12047503).

# A  Symmetric NQS

## A.1  Translation-invariant RBM

Starting from the regular RBM expression

$$\text{RBM}_{\boldsymbol{\theta}}(\boldsymbol{\sigma}) = \exp\left(\sum_{i=1}^{N} a_i \sigma_i\right) \prod_{j=1}^{M} \cosh\left(\sum_{i=1}^{N} W_{ji}\sigma_i + b_j\right), \tag{A.1}$$

symmetries are imposed in the following way: Consider a symmetry group $G$ with elements $g \in G$ acting on the Hilbert space by $T_g |\boldsymbol{\sigma}\rangle = |g\boldsymbol{\sigma}\rangle$, where $T_g$ is a linear unitary operator representing the action of $g$ on the spins and $g\boldsymbol{\sigma}$ denotes the permuted spin configuration. Every $g \in G$ induces a permutation $\pi_g$ of the spins as $(g\boldsymbol{\sigma})_i = \sigma_{\pi_{g^{-1}}(i)}$, where $\sigma_i$ denotes the $i$-th spin on the lattice. After introducing the hidden feature density $\alpha = M/B \in \mathbb{N}$, with $B = N/|G|$ being the number of basis spins such that any other spin is connected to one basis spin by some permutation $\pi_g$, the symmetrized RBM architecture is given by [31]

$$\text{RBM}_{\boldsymbol{\theta}}^{\text{symm}}(\boldsymbol{\sigma}) = \exp\left(\sum_{b=1}^{B} a_b \sum_{g \in G} \sigma_{\pi_g(b)}\right) \prod_{g \in G} \prod_{j=1}^{\alpha B} \cosh\left(\sum_{i=1}^{N} W_{ji}\sigma_{\pi_g(i)} + b_j\right). \tag{A.2}$$

The number of hidden units $M = \alpha B$ is chosen such that this expression reduces back to equation A.1 for the trivial group $G = \{e\}$. Notice, for fixed $\alpha$, the number of trainable parameters is reduced by the factor of $|G|$ in comparison to the regular RBM, although the cost of evaluating the wave function for a single input stays the same. In this work, we include just translational symmetries for which $|G| \sim O(N)$. The checkerboard model is defined on the three-dimensional cubic lattice but it is only symmetric under translations by two lattice sites in any direction due to the checkerboard-like structure. Hence, this leads to $B = 8$, in comparison to $B = 2$ for the 2d toric code, for instance.

## A.2  Symmetric correlation-enhanced RBM (cRBM)

Motivated by the cRBM architecture for the 2d toric code [50], we include bond and non-contractible loop correlators in addition to cube correlators. The bond correlators $C_i^{\text{bond}}$ contain the pairwise products of nearest-neighbor spins. The values of the non-contractible loop correlators $C_i^{\mu-\text{loop}}$ with $\mu \in \{x, y, z\}$ inform the network about the different ground state sectors, as some might be energetically favourable over others in the presence of magnetic fields due to finite-size effects.

Moreover, we include additional hidden units that are just connected to the non-contractible loop correlators, allowing the network to separately adjust wave function amplitudes only in terms of the ground state sectors. Similar to the regular RBM, we symmetrize

this Ansatz by sharing weights over the configurations connected by translations. This is possible since symmetries of the system correspond to graph-automorphisms of the lattice, thereby transforming correlators of one type into each other. For instance, any translation transforms a cube into another cube, hence inducing a permutation on the cube correlators. Hence, our cRBM Ansatz for the checkerboard model reads

$$
\text{cRBM}_{\boldsymbol{\theta}}(\boldsymbol{\sigma})^{\text{symm}} = \exp\left( a \sum_i \sigma_i \right) \exp\left( a^{\text{bond}} \sum_i C_i^{\text{bond}} \right) \exp\left( a^{\text{cube}} \sum_i C_i^{\text{cube}} \right) \tag{A.3}
$$

$$
\times \prod_{j=1}^{\alpha B} \prod_{g \in G} \cosh\left( b_j + \sum_i W_{ji} \sigma_{\pi_g(i)} + \sum_i W_{ji}^{\text{bond}} C_{\pi_g(i)}^{\text{bond}} + \sum_i W_{ji}^{\text{cube}} C_{\pi_g(i)}^{\text{cube}} \right.
$$

$$
\left. + \sum_{\mu \in \{x,y,z\}} \sum_i W_{ji}^{\mu-\text{loop}} C_{\pi_g(i)}^{\mu-\text{loop}} \right)
$$

$$
\times \prod_{\mu \in \{x,y,z\}} \exp\left( a^{\mu-\text{loop}} \sum_i C_i^{\mu-\text{loop}} \right) \cosh\left( b^{\mu-\text{loop}} + \sum_i (W')_i^{\mu-\text{loop}} C_{\pi_g(i)}^{\mu-\text{loop}} \right).
$$

For the hidden feature density, we choose $\alpha = 1/4$, resulting in two hidden features which are shared according to the translational symmetries of the system. This choice is guided by the discussion in Section 4, which highlights that two hidden units per cube are sufficient to parameterize the unperturbed checkerboard model. Notice, in our parametrization the visible biases of the single spins and bonds are shared even over sectors not connected by translations. Due to the enlarged units cell of the checkerboard model, this greatly simplifies the implementation of the network while maintaining translational invariance. We found that sharing the visible biases in this way has no noticeable impact on the performance. For a fixed feature density $\alpha$ and linear system size $L$, this Ansatz contains $9 + 3L^2 + \alpha B(1 + 3L^2 + \frac{9}{2}L^3)$ trainable parameters, with $B = 8$ for translations in the checkerboard model. For the $4 \times 2 \times 2$ system, there are 215 variational parameters.

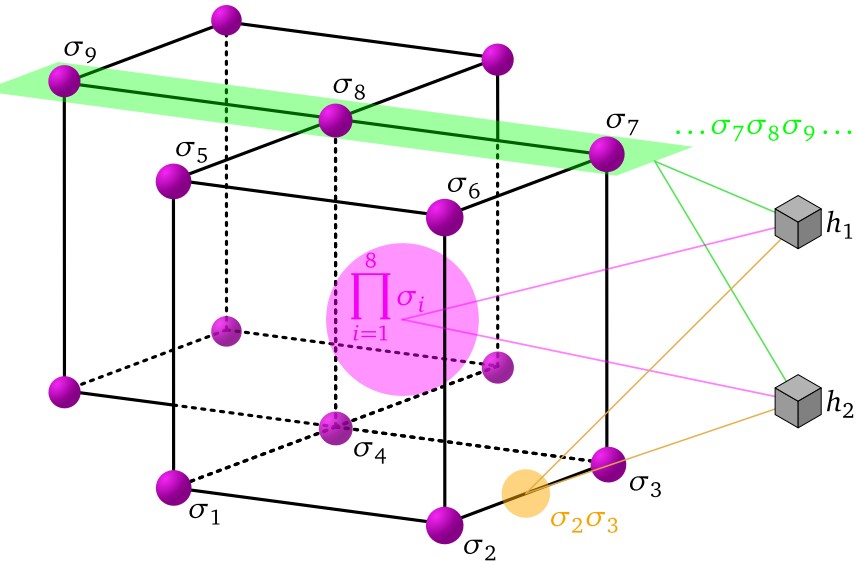

Figure 15: Illustrated is an example of the different correlators included in the cRBM architecture applied to the checkerboard model. A non-contractible loop correlator (green), cube correlator (purple) and bond operator (orange) are constructed from the input configuration and connected to the hidden units (grey).

# B  NQS optimization and transfer learning

The gradient $f_{\theta_j}$ of the variational energy with respect to a real-valued parameter $\theta_j$ can be expressed as

$$f_{\theta_j} = \nabla_{\theta_j} \frac{\langle \psi_{\boldsymbol{\theta}} | H | \psi_{\boldsymbol{\theta}} \rangle}{\langle \psi_{\boldsymbol{\theta}} | \psi_{\boldsymbol{\theta}} \rangle} = 2\mathrm{Re}\left[ \left\langle O_j^\dagger H - \langle H \rangle_{\psi_{\boldsymbol{\theta}}} O_j^\dagger \right\rangle_{\psi_{\boldsymbol{\theta}}} \right], \tag{B.1}$$

where the derivative operators $O_j$ are defined by

$$O_j |\psi_{\boldsymbol{\theta}}\rangle := \sum_{\boldsymbol{\sigma}} \psi_{\boldsymbol{\theta}}(\boldsymbol{\sigma}) \partial_{\theta_j} \log(\psi_{\boldsymbol{\theta}}(\boldsymbol{\sigma})) |\boldsymbol{\sigma}\rangle \Leftrightarrow \langle \boldsymbol{\sigma} | O_j | \boldsymbol{\sigma}' \rangle = \delta_{\boldsymbol{\sigma}\boldsymbol{\sigma}'} \partial_{\theta_j} \log(\psi_{\boldsymbol{\theta}}(\boldsymbol{\sigma})) = \delta_{\boldsymbol{\sigma}\boldsymbol{\sigma}'} O_j(\boldsymbol{\sigma}). \tag{B.2}$$

If $\psi_{\boldsymbol{\theta}}$ is holomorphic in terms of its complex-valued parameter $\theta_j$, which is the case for the RBM architectures presented in this work, we instead arrive at the following expression for the gradient:

$$f_{\theta_j} = \left\langle O_j^\dagger H - \langle H \rangle_{\psi_{\boldsymbol{\theta}}} O_j^\dagger \right\rangle_{\psi_{\boldsymbol{\theta}}} = \mathrm{Cov}(O_j, H). \tag{B.3}$$

Otherwise, we can simply treat any complex-valued parameter as two independent real-valued parameters and again apply Equation B.1. Finally, this expression can be computed by sampling configurations $\boldsymbol{\sigma}_i$ from the Born distribution:

$$\mathrm{Cov}(O_j, H) = \left\langle O_j^\dagger H - \langle H \rangle_{\psi_{\boldsymbol{\theta}}} O_j^\dagger \right\rangle_{\psi_{\boldsymbol{\theta}}} \approx \frac{1}{N_{\mathrm{samples}}} \sum_{i=1}^{N_{\mathrm{samples}}} \left( O_j^*(\boldsymbol{\sigma}_i) E_{\mathrm{loc}}(\boldsymbol{\sigma}_i) - \langle E_{\mathrm{loc}} \rangle O_j^*(\boldsymbol{\sigma}_i) \right), \tag{B.4}$$

where the so-called local estimators of $H$ are defined as

$$E_{\mathrm{loc}}(\boldsymbol{\sigma}) := \sum_{\boldsymbol{\eta}} \langle \boldsymbol{\sigma} | \mathcal{O} | \boldsymbol{\eta} \rangle \frac{\psi_{\boldsymbol{\theta}}(\boldsymbol{\eta})}{\psi_{\boldsymbol{\theta}}(\boldsymbol{\sigma})}. \tag{B.5}$$

Note that the expectation values $\mathrm{Cov}(O_j, H)$, which are also referred to as forces, can be conveniently expressed as a vector-Jacobian product (VJP) in the following way:

$$\mathrm{Cov}(O, H)^\dagger = \boldsymbol{v}^\dagger J, \tag{B.6}$$

where $v_i := N_{\mathrm{samples}}^{-1}(E_{\mathrm{loc}}(\boldsymbol{\sigma}_i) - \langle E_{\mathrm{loc}} \rangle)$ and $J_{ij} = O_j(\boldsymbol{\sigma}_i)$ is the Jacobian of the map

$$\left( \boldsymbol{\sigma}_1, ..., \boldsymbol{\sigma}_{N_{\mathrm{samples}}} \right) \longmapsto \left( \log \psi_{\boldsymbol{\theta}}(\boldsymbol{\sigma}_1), ..., \log \psi_{\boldsymbol{\theta}}(\boldsymbol{\sigma}_{N_{\mathrm{samples}}}) \right). \tag{B.7}$$

This is useful because the local estimators can now be computed efficiently by means of automatic differentiation.

Markov chain Monte Carlo methods are typically deployed to obtain the samples $\boldsymbol{\sigma}_i$. We make use of the Metropolis-Hastings algorithm, which only requires access to a function proportional to the target probability distribution in order to make sampling from the unnormalized NQS feasible. Our update rule includes single spin-flips, cube-flips and flips of non-contractible loops along all three spatial directions (defined precisely like the $\sigma^z$-type logical operators) with probabilities 0.51, 0.25, 0.08, 0.08, 0.08, respectively. This improves the acceptance rate and mixing of the chains, in particular deep in the fracton phase.

In order to minimize the variational energy of the NQS, we make use of stochastic reconfiguration, a modified gradient-descent update introduced by [54]. Stochastic reconfiguration (SR) can be derived as a second-order method that tries to align the parameter update with imaginary time evolution and is the standard optimization technique for NQS applied to the ground state problem. With SR, the parameter update rule reads

$$\boldsymbol{\theta}_{\mathrm{new}} = \boldsymbol{\theta}_{\mathrm{old}} - \gamma S^{-1} f_{\boldsymbol{\theta}_{\mathrm{old}}}, \tag{B.8}$$

where the quantum geometric tensor (QGT) $S$ is defined as

$$S_{ij} = \text{Cov}(O_i, O_j) = \langle O_i^\dagger O_j \rangle_{\psi_\theta} - \langle O_i^\dagger \rangle_{\psi_\theta} \langle O_j \rangle_{\psi_\theta}. \tag{B.9}$$

Alternatively, we can write

$$S = \overline{O}^\dagger \overline{O}, \qquad \text{where} \quad \overline{O}_{ij} := \frac{1}{\sqrt{N_{\text{samples}}}} \left( O_j(\sigma_i) - \langle O_j \rangle \right). \tag{B.10}$$

The introduced hyperparameter $\gamma \in \mathbb{R}_+$ is called learning rate. Since $S$ is only a noisy estimate obtained from MC samples and is usually not directly invertible, we regularize it by applying a diagonal shift $S \mapsto S + \epsilon^{\text{diag}} \mathbb{1}$. Then, we use the iterative conjugate gradient method for the inversion. The parameter update B.8 is then applied repeatedly for $N_{\text{iter}}$ iterations. Algorithm 1 depicts all essential elements of a training loop for NQS.

The implementation of the transfer learning protocol is straightforward: First, save the optimized parameters $\theta^*_{\text{prior}}$ obtained by training the NQS for some field configuration $\vec{h}_{\text{prior}}$. Moreover, we save the final $N_{\text{chains}}$ configurations of the Markov chains $\vec{s}^{(N_{\text{epochs}})}_{\text{prior}}$. Then, simply use these parameters and chain states as the new initial parameters $\theta_0$ and chain states $\vec{s}^{(0)}_{\text{next}}$ when training the NQS for the next field configuration $\vec{h}_{\text{next}}$. For the first NQS training run, we sample the initial parameters from a zero-mean Gaussian with a standard deviation of $\sigma \sim 10^{-2}$. A too large difference $\Delta_h = |\vec{h}_{\text{prior}} - \vec{h}_{\text{next}}|$ between successive field strengths might result in diverging gradients or overflow errors. In general, we found the performance during transfer learning robust for values $\Delta_h \lesssim 0.1$. Of course, $\Delta_h$ can be further reduced to achieve higher resolution in the critical region, for instance. This needs to be weighed against the increased computational cost, as transfer learning does not allow the training of NQS for different field strengths in parallel.

While this optimization scheme is derived from numerical experiments and basis considerations, it can also be seen as an implementation of variational neural annealing (VNA) [60].

---

**Algorithm 1:** NQS optimization.

**Data:** neural network wave function $\psi_\theta$, Hamiltonian $H$, MCMC sampler including update rule, hyperparameters

**Result:** optimized parameters $\theta^*$

1   initialize network parameters $\theta_0$;

2   initialize MCMC sampler configurations $\vec{s}^{(0)} \equiv (s_l^{(0)})_{l \in \{1, ..., N_{\text{chains}}\}}$;

3   $e \leftarrow 0$;

4   **while** $e < N_{\text{iter}}$ **do**

5     $(\sigma_i)_{i \in \{1, ..., N_{\text{samples}}\}}, \vec{s}^{(e+1)} \leftarrow \text{sampler}(\psi_{\theta_e}, \vec{s}^{(e)})$;     ▷ incl. thermalization

6     $c_{ij} \leftarrow \eta_j$ s.t. $\langle \sigma_i | H | \eta_j \rangle \neq 0$, $j \in \{1, ..., K\}$; ▷ get all $K$ connected states

7     $m_{ij} \leftarrow \langle \sigma_i | H | \eta_j \rangle$;                   ▷ get corresponding matrix elements

8     $E_{\text{loc}}(\sigma_i) \leftarrow \sum_{j=1}^{K} m_{ij} \frac{\psi_\theta(c_{ij})}{\psi_\theta(\sigma_i)}$;           ▷ get all local estimators

9     $\overline{H} \leftarrow \frac{1}{N_{\text{samples}}} \sum_i E_{\text{loc}}(\sigma_i)$;

10    $\mathbf{v} \leftarrow \frac{1}{N_{\text{samples}}} (E_{\text{loc}}(\sigma_i) - \overline{H})$;

11    $f_\theta \leftarrow \text{VJP}(\mathbf{v})$;                         ▷ compute gradients

12    $\Delta\theta \leftarrow S^{-1} f_\theta$;                 ▷ stochastic reconfiguration

13    $\theta_{e+1} = \theta_e - \gamma \Delta\theta$;

14    $e \leftarrow e + 1$;

15   **return** $\theta^* = \theta_{N_{\text{iter}}}$

---

This is a framework in which variational quantum annealing is emulated using variational methods such as NQS in order to find the ground state of some Hamiltonian $H_{\text{target}}$. To this end, an additional driving term $H_{\text{D}}$ is introduced, such that the full Hamiltonian reads $H(t) = H_{\text{target}} + f(t)H_{\text{D}}$, where $f(t)$ describes a time dependent coupling. For simplicity, we fix $f(0) = f(t_0) = 1$ and $f(1) = f(t_f) = 0$. The key idea is as follows: $H_{\text{D}}$ is chosen such that the ground state of $H(t = 0)$ is easy to learn. The NQS is trained to approximate the ground state at $t = 0$, which serves as the initialization for the VNA procedure: First, the time is increased by a small increment $t_{i+1} = t_i + \delta t$. Then, the neural network is trained for some number of gradient-descent (SR) steps to approximate the new instantaneous ground state. Repeating this process until some final time $t_f$ is reached such that $f(t_f) = 0$ should lead to the ground state of $H_{\text{target}}$. At each step, the training of the network at time $t_{i+1}$ is always initialized with the trained parameters and samples from the previous time $t_i$ to ensure adiabaticity. See [60] for a full discussion and more details. Finally, by identifying the target Hamiltonian with $H_{\text{Fracton}}$, the driving term with the Pauli operators $H_{\text{D}} = \sum_i \vec{\sigma}_i$, and the magnetic field strengths $\vec{h}(t)$ with a suitably chosen schedule of the coupling strength $f(t)$, we recover the optimization scheme discussed above.

## C  Complexity

### C.1  Hyperparameters

Table 1 shows the hyperparameters that we used to train the cRBM on the checkerboard model subject to magnetic fields. By making use of one or multiple GPUs, we can efficiently sample from multiple Markov chains in parallel, denoted by $N_{\text{chains}}$. New samples are obtained after $N_{\text{updates}}$ update steps in the Metropolis-Hastings algorithm to reduce their auto-correlation. Each chain is individually thermalized after every parameter change by omitting the first $N_{\text{therm}}$ samples. The zero-variance principle only applies to the energy and not to other observables like the magnetization. In order to still obtain accurate estimates of these observables, a much larger number of samples is required, which we denote as $N_{\text{expect}}$. Both the learning rate $\gamma$ and the diagonal regularization $\epsilon^{\text{diag}}$ of the quantum geometric tensor (QGT) are gradually reduced after a couple hundred training iterations. We use a smaller learning rate for left-right sweeps in order to stabilize the training. The reduced number of samples for training on the $L = 8$ lattice is due to computational constraints. With these hyperparameter choices, the time required to train the NQS on an A100 GPU for a single magnetization takes about 2.5h and 6h for $L = 6$ and $L = 8$, respectively (not including the evaluation using $N_{\text{expect}}$ samples). Using two V100 GPUs, it takes about 30min to train for a single field configuration on an $L = 4$ lattice. In total, training for all field configurations and system sizes required about 140 V100-hours and 390 A100-hours (without evaluation).

### C.2  Complexity and parallelization

Table 2 summarizes dominant contributions to different steps of the NQS optimization process in terms of time and memory complexity. Sampling with the Metropolis-Hastings algorithm requires a large number of network evaluations $\sim O(N^2)$; there are $N_{\text{updates}} \sim N$ update steps to obtain a new sample, each one requiring a new network evaluation with cost $F$. For a $G$-symmetric RBM architecture with dense filters, it holds that $F \sim N|G|$. For any network architecture, we have that $F$ is at least $O(N)$ in order to process the whole input configuration of size $N$. For some network architectures the sampling cost can be reduced, for instance by using look-up tables for RBM architectures. Still, sampling with the Metropolis-Hastings algorithm belongs to the most time consuming steps in NQS optimization. For every sample,

there are typically $K \sim N$ connected matrix elements that must be calculated according to the specific Hamiltonian. The time scaling in $N$ stems from the fact that each connected element is a vector of length $N$ that must be constructed and written to memory, but this is generally a cheap operation. Storing the connected elements is typically the most memory expensive task, however, it is possible to batch the calculations of the local estimators such that not all connected elements must be stored at the same time. The automatic differentiation capabilities of JAX allow for the calculation of VJPs at a cost that scales linearly with the evaluation cost of the network on all samples. However, reverse-mode automatic differentiation requires saving intermediate results, which are calculated during the forward pass of the network. We denote the corresponding memory requirement by $O(F)$ (in the memory context); note that this is not the same amount of (and usually less than the) memory required for storing the parameters $P$ of the network. Stochastic reconfiguration, in principle, requires the inversion of a $P \times P$ matrix. This scales as $O(P^3)$, thereby hindering the deployment of large neural networks. The iterative conjugate gradient (CG) method can be used to invert the QGT approximately, but it a requires suitable regularization and a potentially large number of iterations if the QGT has a high condition number. Note that at no point in time the full QGT needs to be constructed in memory. Instead, its action on a vector can be recalculated from the Jacobian of the network (Jacobian-dense) or on-the-fly, we refer to the Netket documentation for details [56]. Recently, [68, 69] proposed a modified algorithm to perform SR based on the reduced QGT, which only has $N_{\text{samples}} \times N_{\text{samples}}$ entries. This enables SR even for large neural networks and also allows for direct inversions if the number of samples is not too large.

Since the proposed transfer learning protocol does not allow the training of NQS for multiple field strengths in parallel, it is crucial to optimize the training of individual NQS as much as possible. In fact, Algorithm 1 can be parallelized to a large degree, as is illustrated in Figure 16. In particular the sampling, the calculation of local estimators, the computation of the VJP for the force vector, and the application of $S = \overline{O}^\dagger \overline{O}$ to some vector $\boldsymbol{p}$ (required for the CG solver) can be parallelized along the sample dimension. However, this speed-up is only noticeable for a large number of samples because a single GPU can already parallelize many computations, like sampling from $O(100)$ Markov chains in parallel in the same time as sampling from a single chain, and communication between ranks causes additional overhead.

Table 1: Hyperparameters for the checkerboard model. The $N_{\text{samples}}$ samples are evenly distributed over all $N_{\text{chains}}$ chains, each chain is thermalized individually. A sample is obtained after performing $N_{\text{updates}}$ update steps in the Metropolis algorithm. $N_{\text{expect}}$ denotes the total number of samples used to evaluate physical observables. The sampling parameters apply only to larger system sizes ($L \geq 4$) where full summation is not available.

| Hyperparameter | $4 \times 2 \times 2$ | $L = 4$ | $L = 6$ | $L = 8$ |
|---|---|---|---|---|
| $N_{\text{samples}}$ | - | $2^{14}$ | $2^{14}$ | $2^{12}$ |
| $N_{\text{chains}}$ | - | 1024 | 1024 | 1024 |
| $N_{\text{updates}}$ | - | $4^3$ | $6^3$ | $8^3$ |
| $N_{\text{therm}}$ | - | 24 | 24 | 20 |
| $N_{\text{expect}}$ | - | $24 \times N_{\text{samples}}$ | $24 \times N_{\text{samples}}$ | $96 \times N_{\text{samples}}$ |
| $N_{\text{iter}}$ | 1200 | 1200 | 1200 | 1500 |
| $\gamma$ (right-left) | $10^{-2} \to 10^{-3}$ | $10^{-2} \to 10^{-3}$ | $10^{-2} \to 10^{-3}$ | $3 \times 10^{-3} \to 10^{-3}$ |
| $\gamma$ (left-right) | $10^{-2} \to 10^{-3}$ | $3 \times 10^{-3} \to 10^{-3}$ | $3 \times 10^{-3} \to 10^{-3}$ | $3 \times 10^{-3} \to 10^{-3}$ |
| $\epsilon^{\text{diag}}$ | $10^{-4} \to 10^{-5}$ | $10^{-4} \to 10^{-5}$ | $10^{-4} \to 10^{-5}$ | $10^{-4} \to 10^{-5}$ |

Table 2: Overview of time and memory complexities of individual steps during NQS optimization. $N$ denotes the number of qubits, $K$ denotes the maximum number of connected elements for any input state. $F$ denotes the cost of evaluating the network for a single input in the time context and the space required to store intermediate values during the forward pass in the memory context. $P$ denotes the number of variational parameters in the network.

| | time complexity | memory complexity |
|---|---|---|
| sampling | $O\big((N_{\text{therm}} + N_{\text{samples}})N_{\text{updates}}F\big)$ | $O(N_{\text{samples}}N)$ |
| connected matrix elements | $O(N_{\text{samples}}KN)$ | $O(N_{\text{samples}}KN)$ |
| local estimators | $O(N_{\text{samples}}KF)$ | $O(N_{\text{samples}}NK)$ |
| forces | $O(4N_{\text{samples}}F)$ | $O(N_{\text{samples}}F)$ |
| stochastic reconfiguration | direct: $O(P^3)$ <br> CG: $O(PN_{\text{samples}}N_{\text{iter}})$ | direct: $O(P^2)$ <br> Jacobian-dense: $O(PN_{\text{samples}})$ <br> on-the-fly: $O(FN_{\text{samples}})$ |

### C.3 Code implementation

Our code [51] is based on the Netket library [56] and JAX [57]. JAX is a machine learning and HPC framework written in Python that supports automatic differentiation and functional transformations on CPU, GPU and TPU. Jax traces the action of functions on their input to construct the computational graph, which is then compiled using XLA, an open-source compiler for machine learning applications. For instance, automatic differentiation allows for efficient computation of the forces for the NQS optimization through vector-Jacobian products, and functional transformations such as `vmap` make it simple to sample from multiple $O(100)$ Markov chains in parallel in the same time as a single chain by making use of the parallel compute capabilities of GPUs. We implemented the NQS architectures, the update rule for the Metropolis sampler, and the operators ourselves. For the basic infrastructure, like computing gradients and SR, we rely on the Netket library.

To demonstrate an advantage of our implementation, we briefly discuss the implementation of the Hamiltonians. It is generally not tractable to store any operator like the Hamiltonian as a matrix for large system sizes. Instead, we implement operators as functions that take as input some spin configuration $|\boldsymbol{\sigma}\rangle$ and return the connected elements $|\boldsymbol{\eta}\rangle_i$ such that $\langle\sigma|H|\eta_i\rangle \neq 0 \ \forall i$, as well as the corresponding matrix elements $\langle\sigma|H|\eta_i\rangle$. Netket's built-in local operator API makes it very simple to implement operators that can be written as a sum of local operators. However, in order to make this API so flexible, it works with matrix representations of the individual local operators behind the scenes. For the checkerboard model, Netket handles $O(N)$ different matrices of size $2^8 \times 2^8$ corresponding to the stabilizer generators defined on the cubes, which leaves much room for performance improvements.

Here, we compare the performance between our custom operator implementation and the NetKet local operator version in Figure 17 in the presence of a magnetic field along the $x$-direction. For the Checkerboard model, every input configuration leads to $1 + L^3/2 + L^3$ connected elements (1 diagonal element, $L^3/2$ from the $A_\mathcal{C}$ cube operators and $L^3 = N$ from the magnetic field) and corresponding matrix elements. All computations are run on a single Nvidia V100 GPU. The python script to reproduce these plots is located in our repository [51] under the name `custom_operator_performance.py`.

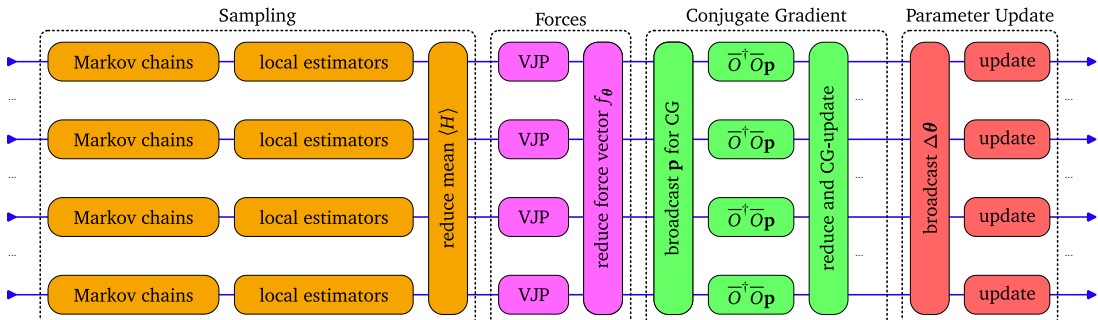

Figure 16: Visualization of an NQS optimization step distributed over multiple hosts/processes, in this example four. Most communications between ranks are required for the iterative CG procedure. We typically run $O(100)$ chains in parallel on each rank. For more detailed benchmarks investigating the speed-up when working with multiple hosts, see [56].

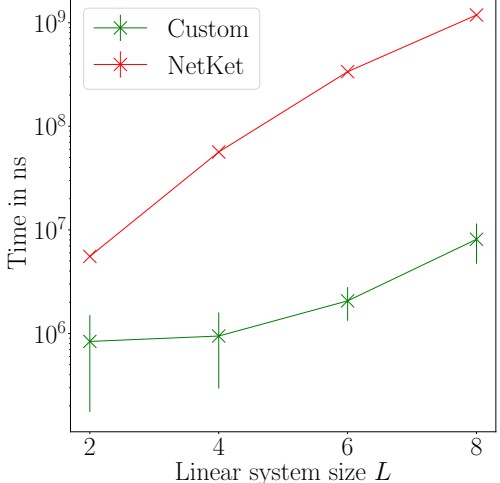

| Size $L$ | Custom Operator | NetKet Operator |
|----------|-----------------|------------------|
| 2 | 8e+05 | 5.54e+06 |
| 4 | 9e+05 | 5.65e+07 |
| 6 | 2.1e+06 | 3.36e+08 |
| 8 | 8.1e+06 | 1.186e+09 |

Figure 17: The time required to compute connected elements and corresponding matrix elements for the checkerboard model subject to a magnetic field in x-direction is shown for NetKets built-in local operator interface (red) and our custom implementation (green) in nanoseconds for different system sizes $L \times L \times L$. 2048 samples are used.

# D  Convergence diagnostics

Figure 18 and 19 show different MC statistics for the $L = 4$ and $L = 8$ checkerboard model, respectively, averaged over the last 200 training iterations. Next to the acceptance ratio of the weighted update rule and the estimated auto-correlation time $\tau$, they also display the split-$\hat{R}$ value and the so-called V-score. The split-$\hat{R}$ value, introduced by [58], is a modified version of the original $\hat{R}$ value [71] that measures the ratio between the intra-chain and inter-chain variance when sampling from multiple chains in parallel. High $\hat{R}$ values indicate slow mixing of the chains, which means that the variance between chains exceeds the variance within any single chain. Ideally, $\hat{R}$ should be close to 1 and $\hat{R} \lesssim 1.1$ is deemed acceptable. For the split-$\hat{R}$ value, each chain is split in half and treated independently while applying the same analysis as for the regular $\hat{R}$, resulting in higher sensitivity to non-convergence of single chains. The V-score [59] is a dimensionless intensive number defined by $N\frac{\text{Var}(E)}{E^2}$, which can be used to compare the variational accuracy of different methods on the same ground state problem; we refer to the original paper for a detailed discussion.

Although Figure 18 and 19 confirm that the $L = 8$ problem is harder in terms of sampling and V-score, the acceptance rate of updates does not fall significantly below 0.5 and the auto-correlation times stay sufficiently short. Unsurprisingly, the critical region is particularly challenging for the NQS. This is indicated by an increase of the split-$\hat{R}$ value and the V-score, as well as the auto-correlation times, although the latter stay very low overall; in particular, the high split-$\hat{R}$ values close to the critical point seems problematic at first glance. As mentioned earlier, split-$\hat{R}$ is particularly sensitive to non-convergence of individual chains and might not be the most suitable score for massively parallel sampling with $O(1000)$ chains or more [58]. Close to the first-order transition, we expect the Markov chains to remain in different (meta-)stable configurations; even after passing the transition point, the variational state and the chains tend to remain in the (meta-stable) phase in which the state was initialized. This is precisely what causes the observed hysteresis. As the chains begin to explore the region of Hilbert space corresponding to the other phase, the variance increases significantly. At some point, the NQS and the majority of the Markov chains transition to the new phase in order to decrease the variational energy. (This is not observed for left-right sweeps and $L = 6, 8$, likely due to slow mixing and the stability of the fracton phase for larger system sizes.) We observe that about $O(10)$ Markov chains remain in the initial meta-stable phase even after the NQS transitioned towards the new phase. However, their impact on estimated observables should not be significant in the face of the $O(1000)$ parallel chains used for sampling, but they still cause increased variance (lower than in the critical region, higher than deep in the initial phase) and split-$\hat{R}$. Hence, it can be argued that the increased variance and split-$\hat{R}$ in the critical region is a natural consequence of the first-order transition, while the increased variance after transitioning to the new phase is a sampling-related issue. More work needs to be done - like improved sampling techniques, a denser grid of field values, or even larger networks - in order to make precise statements about the exact location of the transition point, but the nature and rough location of the transition should have been captured well by our methods.

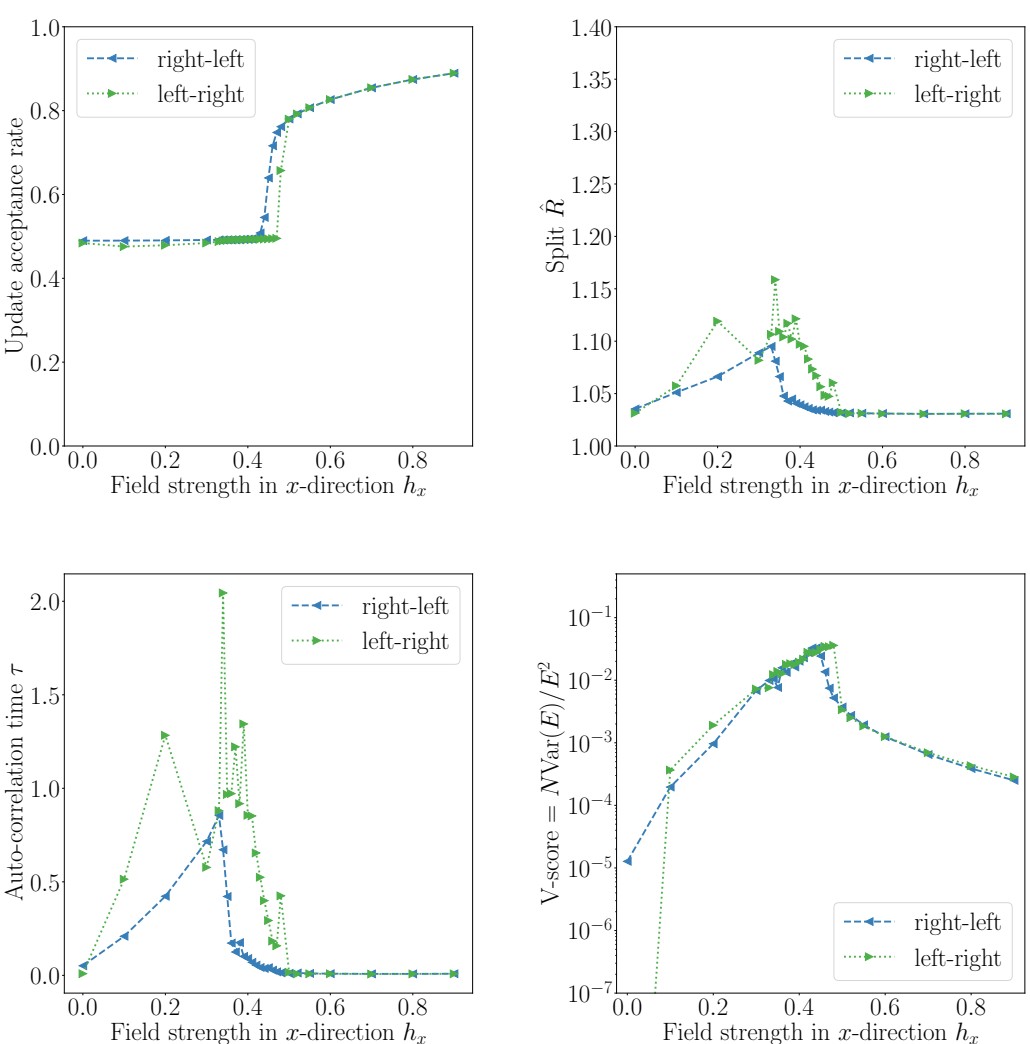

Figure 18: Different MCMC diagnostics averaged over the last 200 training iterations to assess the quality of the sampling process for the $L = 4$ Checkerboard model and varying magnetic $x$-fields are shown.

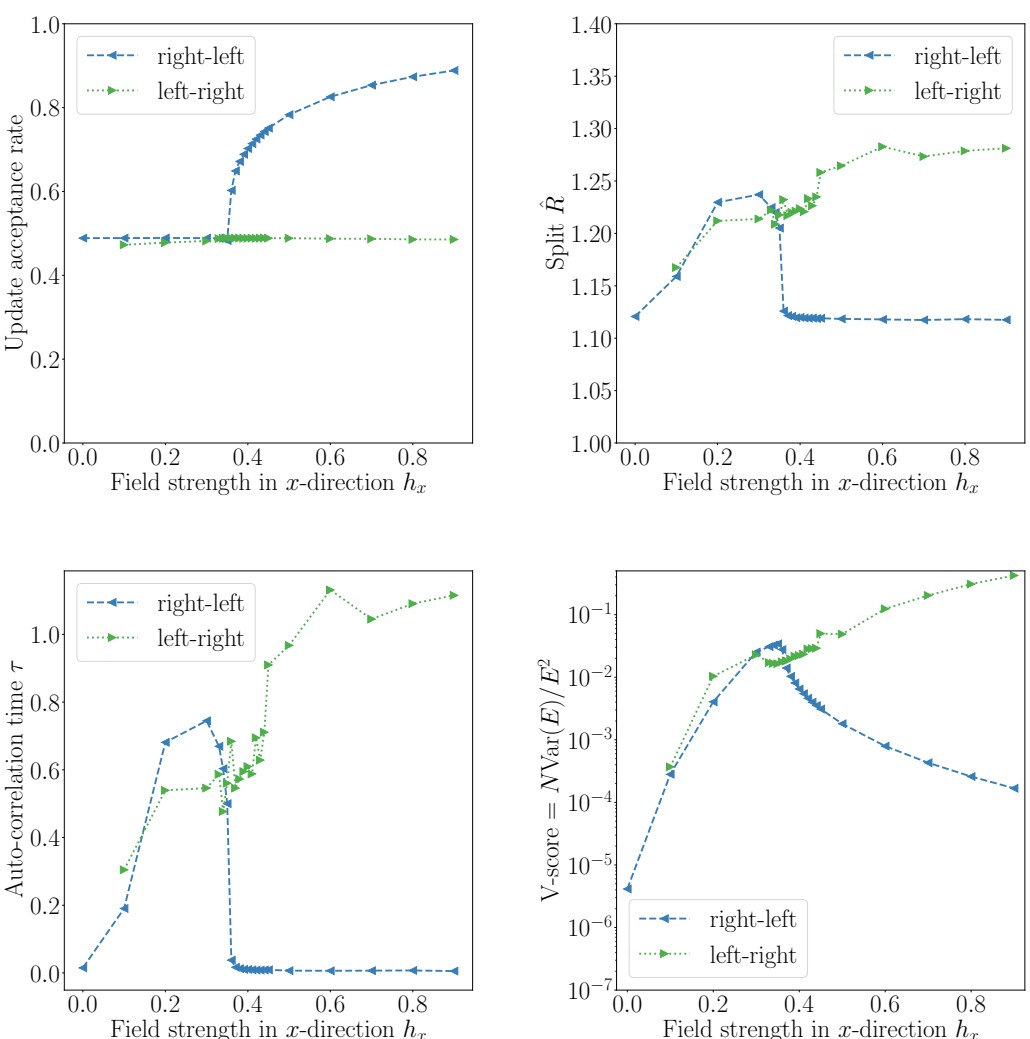

Figure 19: Different MCMC diagnostics averaged over the last 200 training iterations to assess the quality of the sampling process for the $L = 8$ Checkerboard model and varying magnetic $x$-fields are shown.

# E Supplementary results

## E.1 Hyperparameter comparison

(a) $\sigma = 0.01$, lr $= 0.01$.

(b) $\sigma = 0.01$, lr $= 0.1$.

(c) $\sigma = 0.1$, lr $= 0.01$.

(d) $\sigma = 0.1$, lr $= 0.1$.

Figure 20: Comparison of different model architectures for learning the ground state of the pure Checkerboard model on a $4 \times 2 \times 2$ lattice. The real and imaginary parts of the parameters are initialized separately using a centered uniform distribution with given standard deviations. The number in parentheses corresponds to the number of hidden units. The parameter counts for the different models are 208 for the FFNN, 288/152 for the RBM(16)/(8), 137/69 for the symmetric RBM(8)/(4), 88 for the symmetric NN, 120 for the Jastrow and 52 for the cRBM architecture. The ground state energy is indicated by the dashed grey line.

## E.2 X-cube model

In full analogy to the checkerboard simulations, we have included star, loop and bond correlators into the cRBM parametrization. Moreover, cube flips are proposed with a small probability at each update step during sampling. In order to stabilize the computations, a denser grid for the field strengths and slightly stronger diagonal regularization ($\epsilon^{\text{diag}} = 10^{-2} \to 10^{-3}$) were required. We use $2^{13}$ samples during training for $L = 3, 4$ and $2^{12}$ samples for $L = 5$, and the number of stochastic reconfiguration steps after each field change are adapted dynamically. We choose a feature density of $\alpha = 1$, leading to 3 hidden units. Otherwise, the choice of hyperparameters is similar to Table 1. The networks contain 1591, 3673, 7075 parameters for $L = 3, 4, 5$, respectively.

Fig. 21 shows the energy per site and magnetization in the X-cube model subject to a magnetic field in x-direction for different system sizes. The $L = 3$ lattice is too small to exhibit a clear first-order transition, while the larger lattices display a strong hysteresis. We observe a strong first-order transition at $h_{x,c} \approx 0.91(1)$, determined from the intersection point of the energy curves for the largest system size in Fig. 21, which is in accordance with existing results: Previously, the critical value for the magnetic field in x-direction was estimated to be $h_{x,c} \approx 0.9$ [44] via quantum Monte Carlo simulations, and $h_{x,c} = 0.9196 \pm 0.0012$ [46] from high-order series expansions.

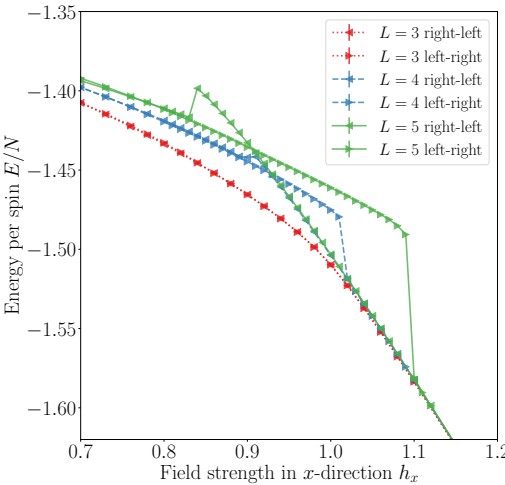
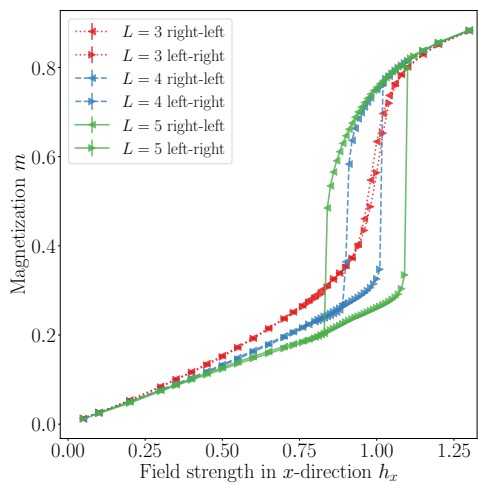

Figure 21: Comparison of the energy per spin and magnetization obtained from the symmetric cRBM architecture trained on the X-cube model for different system sizes and sweep directions. The total number of qubits in the system is equal to $3L^3$.

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
