# Peer review of "Neural Quantum State Study of Fracton Models"

_SciPost Physics, doi:SciPost Phys. 18, 112 (2025)_

## Round 1 · Referee Report · Anonymous (Referee 1) · 2025-2-4

Report

I think the updated manuscript is suitable for publication and the presented study is substantial enough for SciPost Physics journal.

Recommendation

Publish (meets expectations and criteria for this Journal)

---

## Round 1 · Referee Report · Anonymous (Referee 2) · 2025-2-27

Report

I appreciate the authors' detailed responses and clarifications, particularly regarding the role of hysteresis, transfer learning, and visible biases. After reviewing their explanations and the revised manuscript, I realized that I had some misconceptions about certain aspects of their work. The authors’ clarifications have significantly enhanced my understanding of their approach and results.

The revisions, including the detailed responses, additional figures, and supplemental materials, have thoroughly addressed and refuted all my major concerns. I believe these improvements have strengthened the manuscript, making it a valuable contribution to the field.

I, therefore, recommend the paper for publication in SciPost Physics.

Recommendation

Publish (meets expectations and criteria for this Journal)

---

## Round 1 · Author Response

We thank the referees for their valuable comments, suggestions and questions, which
have led to a greatly enhanced manuscript.
We understand that the main concern of referee 2 is missing proof that existing
results can be reliably replicated with our method, which makes the newly achieved
results for the Checkerboard model less trustworthy. Referee 1 raises important
questions regarding the nature of the phase transition in conjunction with the employed
transfer learning protocol, and makes valuable suggestions for a better comparison of
the cRBM architecture against other networks under different hyperparameter settings.
Here, we briefly explain how the revised Manuscript improves on these aspects. For
the detailed response to all suggestions / question raised by the referees, we would
like to refer to our replies to the referee reports on the original submission.
Most notably, we performed intensive additional simulations on the X-cube model
on up to 375 qubits (L=5) for finite x-fields. We have been able to replicate the
hysteresis / first-order transition (hc ≈ 0.91) which was previously investigated
through quantum Monte Carlo simulations (hc ≈ 0.9 Ref.[44]) and high-order series
expansions (hc ≈ 0.92 Ref.[46]). We think these results underline the physicality of
the hysteresis detected by our method and, in conjunction with our original results for
the checkerboard model, they should demonstrate NQS as a new ”workhorse” method
for such systems. We thank referee 2 for their feedback which lead to this valuable
addition to our manuscript, which hopefully lifts the main concerns of Referee 2 and
addresses doubts raised by referee 1 about the meaning of the hysteresis.
Thanks to referee 1’s detailed suggestions, we have substantially expanded the
content of Figure 9 and added a Figure 20, thereby providing a more extensive perfor-
mance comparison of different model architectures under different hyperparameter
settings. We have included results for normal and uniform initialization, different
combinations of learning rate and standard deviation of the initial parameters, and
for the Jastrow wave function and more RBM architectures with different parameter
counts. We hope this better demonstrates that the cRBM displays advantages over
other architectures in terms of convergence speed, hyperparameter robustness and
parameter efficiency.
Finally, we want to thank referee 1 for pointing attention to different parts of the
original manuscript which could be improved or need further explanation. We have
addressed all questions in great detail in the replies to the original referee reports, and
reworked several parts of the manuscript by rephrasing or adding additional references
/ explanations accordingly. This includes the proposed ”there-and-back” approach, the
meaning and detection of the hysteresis, the estimation and precision of the critical
point, and adding context with regards to other works that follow similar transfer
learning ideas.
We thank both referees again for their extensive feedback and valuable comments
and believe that the revised manuscript is ready for publication in SciPost Physics.

---

## Round 1 · List of Changes

• changed the second paragraph of the introduction to clarify the relation between NQS and VMC
  • reformulated end of the paragraph following equation 10 to clarify how NQS can be characterized as a subclass of variational techniques
  • major rework of Figure 9: inclusion of Jastrow wave function, multiple combinations of learning rate and initial standard deviation, increased parameter counts for (symmetric) RBM architectures, different alpha values for RBM architectures
  • adapted discussion of Figure 9 in Section 6 to account for additional results
  • added Figure 20, similar to Figure 9, into Appendix "Supplementary Results" with the difference of using a uniform distribution parameter initialization
  • added Ref.[55] to the end of section 4.2
  • adapted the value of estimated transition point of the checkerboard model in Sec. 7 to reflect uncertainty better, clarified how it is determined
  • new major simulation results of the X-cube model added to new Appendix E, reproducing existing results; added reference to new X-cube results in section 5
  • added a small discussion at the end of Appendix B on how optimization protocol relates to variational neural annealing
  • added references [60] and[61] to Sec. 6
  • corrected order of affiliations, updated the acknowledgement / funding section
  • some typos

---

## Editorial Decision

published